# EIGNN: Efficient Infinite-Depth Graph Neural Networks

**Juncheng Liu**  **Kenji Kawaguchi**  **Bryan Hooi**  **Yiwei Wang**  **Xiaokui Xiao**
National University of Singapore
{juncheng,kenji,bhooi}@comp.nus.edu.sg
wangyw_seu@foxmail.com, xkxiao@nus.edu.sg

## Abstract

Graph neural networks (GNNs) are widely used for modelling graph-structured data in numerous applications. However, with their inherently finite aggregation layers, existing GNN models may not be able to effectively capture long-range dependencies in the underlying graphs. Motivated by this limitation, we propose a GNN model with infinite depth, which we call Efficient Infinite-Depth Graph Neural Networks (EIGNN), to efficiently capture very long-range dependencies. We theoretically derive a closed-form solution of EIGNN which makes training an infinite-depth GNN model tractable. We then further show that we can achieve more efficient computation for training EIGNN by using eigendecomposition. The empirical results of comprehensive experiments on synthetic and real-world datasets show that EIGNN has a better ability to capture long-range dependencies than recent baselines, and consistently achieves state-of-the-art performance. Furthermore, we show that our model is also more robust against both noise and adversarial perturbations on node features.

## 1 Introduction

Graph-structured data are ubiquitous in the real world. To model and learn from such data, graph representation learning aims to produce meaningful node representations by simultaneously considering the graph topology and node attributes. It has attracted growing interest in recent years, as well as numerous real-world applications [32].

In particular, graph neural networks (GNNs) are a widely used approach for node, edge, and graph prediction tasks. Recently, many GNN models have been proposed (e.g., graph convolutional network [14], graph attention network [26], simple graph convolution [30]). Most modern GNN models follow a "message passing" scheme: they iteratively aggregate the hidden representations of every node with those of the adjacent nodes to generate new hidden representations, where each iteration is parameterized as a neural network layer with learnable weights.

Despite the success existing GNN models achieve on many different scenarios, they lack the ability to capture long-range dependencies. Specifically, for a predefined number of layers $T$, these models cannot capture dependencies with a range longer than $T$-hops away from any given node. A straightforward strategy to capture long-range dependencies is to stack a large number of GNN layers for receiving "messages" from distant nodes. However, existing work has observed poor empirical performance when stacking more than a few layers [16], which has been referred to as oversmoothing. This has been attributed to various reasons, including node representations becoming indistinguishable as depth increases. Besides oversmoothing, GNN models with numerous layers require excessive computational cost in practice since they need to repeatedly propagate representations across many layers. For these two reasons, simply stacking many layers for GNNs is not a suitable way to capture long-range dependencies.

35th Conference on Neural Information Processing Systems (NeurIPS 2021), virtual.

Recently, instead of stacking many layers, several works have been proposed to capture long-range dependencies in graphs. Pei et al. [21] propose a global graph method Geom-GCN which builds structural neighborhoods based on the graph and the embedding space, and uses a bi-level aggregation to capture long-range dependencies. However, as Geom-GCN still only has finite layers, it still fails to capture very long range dependencies.

To model longer range dependencies, Gu et al. [10] propose the implicit graph neural network (IGNN), which can be considered as a GNN model with infinite layers. Thus, IGNN does not suffer from any *a priori* limitation on the range of information it can capture. To achieve this, IGNN generates predictions as the solution to a fixed-point equilibrium equation. Specifically, the solution is obtained by an iterative solver. Moreover, they require additional conditions to ensure well-posedness of the model and use projected gradient descent method to train the model. However, the practical limitations of iterative solvers are widely recognized: they can lack robustness; the generated solution is approximated; and the number of iterations cannot be known in advance [23]. In our experiments, we found that IGNN sometimes experiences non-convergence in its iterative solver. Besides the non-convergence issue, the iterative solver of IGNN can be inefficient as IGNN needs to run it once per forward or backward pass.

Motivated by these limitations, we propose our Efficient Infinite-Depth Graph Neural Network (EIGNN) approach, which can effectively capture very long range dependencies in graphs. Instead of relying on iterative solvers to generate the solution like in IGNN [10], we derive a closed-form solution for EIGNN without additional conditions, avoiding the need for projected gradient descent to train the model. Furthermore, we propose to use eigendecomposition to improve the efficiency of EIGNN, without affecting its accuracy.

The contributions of this work are summarized as follows:

- To capture long-range dependencies, we propose our infinite-depth EIGNN model. To do this, we first define our model as the limit of an infinite sequence of graph convolutions, and theoretically prove its convergence. Then, we derive tractable forward and backward computations for EIGNN.

- We then further derive an eigendecomposition-based approach to improve the computational/memory complexity of our approach, without affecting its accuracy.

- We empirically compare our model to recent baseline GNN models on synthetic and real-world graph datasets. The results show that EIGNN has a better ability to capture very long range dependencies and provides better performance compared with other baselines. Moreover, the empirical results of noise sensitivity experiments demonstrate that EIGNN is more robust against both noise and adversarial perturbations.

**Paper outline** In Section 2, we provide an overview of GNNs, implicit models, and the oversmoothing problem. Section 3 introduces the background and major mathematics symbols used in this paper. In Section 4, we first present the EIGNN model and discuss how to train this infinitely deep model in practice. In addition, we show that with eigendecomposition we can reduce the complexity and achieve more efficient computation for EIGNN. In Section 5, we empirically compare EIGNN with other representative GNN methods.

## 2   Related work

**Graph neural network models** GNN models have been successfully used on various graph related tasks [32, 34, 27, 28]. Pioneered by Graph Convolutional Networks (GCNs) [14], many convolutional GNN models [33, 30, 26, 4, 15, 11, 29] use different aggregation schemes and components (e.g., attention[26], skip connection[33, 4]). They only have a finite number of aggregation layers, which makes them unable to capture long-range dependencies. Recurrent GNN models [24, 9, 17, 6] generally share the same parameters in each aggregation step and potentially allow infinite steps until convergence. However, as pointed out in Gu et al. [10], their sophisticated training process and conservative convergence conditions limit the use of these previous methods in practice. Recently, Gu et al. [10] proposes an implicit graph neural network (IGNN) with infinite depth. IGNN uses an iterative solver for forward and backward computations, and requires additional conditions to ensure the convergence of the iterative method. With infinite depth, they aim to capture very long range dependencies on graphs. Another global method Geom-GCN [21] is also proposed for capturing

long-range dependencies using structural neighborhoods based on the graph and the embedding space. However, as Geom-GCN only has finite layers, it fails to capture very long range dependencies.

**Implicit models**  Implicit networks use implicit hidden layers which are defined through an equilibrium point of an infinite sequence of computation. This makes an implicit network equivalent to a feedforward network with infinite depth. Several works [1, 2, 5, 7] show the potential advantages of implicit models on many applications, e.g., language modeling, image classification, and semantic segmentation. Besides practical applications, Kawaguchi [13] provides a theoretical analysis on the global convergence of deep implicit linear models.

**Oversmoothing**  For capturing long-range dependencies, a straightforward method is to stack more GCN layers. However, Li et al. [16] found that stacking many layers make the learned node representations indistinguishable, which is named the *oversmoothing* phenomenon. To mitigate oversmoothing and allow for deeper GNN models, several empirical and theoretical works [16, 15, 4, 35, 19] have been proposed. However, these models still cannot effectively capture long-range dependencies, as shown in our empirical experiments (see Section 5).

## 3  Preliminaries

Let $\mathcal{G} = (\mathcal{V}, \mathcal{E})$ be an undirected graph with node set $\mathcal{V}$ and edge set $\mathcal{E}$, where the number of nodes $n = |\mathcal{V}|$. In practice, a graph $\mathcal{G}$ can be represented as its adjacency matrix $A \in \mathbb{R}^{n \times n}$ and its node feature matrix $X \in \mathbb{R}^{m \times n}$ where the feature vector of node $i$ is $x_i \in \mathbb{R}^m$. In node classification task, given graph data $(\mathcal{G}, X)$, graph models are required to produce the prediction $\hat{y}_i$ for node $i$ to match the true label $y_i$.

**Simple graph convolution**  The Simple Graph Convolution (SGC) was recently proposed by Wu et al. [30]. The pre-softmax output of SGC of depth $H$ can be written as:

$$f_{SGC}(X, W) = WXS^H \in \mathbb{R}^{m_y \times n}, \tag{1}$$

where $X \in \mathbb{R}^{m \times n}$ is the node feature matrix, $S^H = \prod_{i=1}^{H} S \in \mathbb{R}^{n \times n}$ is the product of $H$ normalized adjacency matrix with added self-loops $S \in \mathbb{R}^{n \times n}$, and $W \in \mathbb{R}^{m_y \times m}$ is the matrix of the trainable weight parameters. After the graph convolution, the softmax operation is applied to obtain the prediction $\hat{y}$.

**Notation**  We use $\otimes$ to represent Kronecker product. $\circ$ is used to represent element-wise product between two matrices with the same shape. For a matrix $V \in \mathbb{R}^{x \times y}$, by stacking all columns, we can get $\text{vec}[V] \in \mathbb{R}^{xy}$ as the vectorized form of $V$. The Frobenius norm of $V$ is denoted as $\|V\|_{\text{F}}$. $\|V\|_2$ denotes the 2-norm. We use $I_n$ to represent the identity matrix with size $n \times n$.

## 4  Efficient Infinite-Depth Graph Neural Networks

To capture long-range dependencies, we propose Efficient Infinite-Depth Graph Neural Networks (EIGNN). The pre-softmax output of EIGNN is defined as:

$$f(X, F, B) = B \left( \lim_{H \to \infty} Z_{X,F}^{(H)} \right), \tag{2}$$

where $F \in \mathbb{R}^{m \times m}$ and $B \in \mathbb{R}^{m_y \times m}$ represent the trainable weight parameters, and $Z^{(H)} = Z_{X,F}^{(H)}$ is the output of the $H$-th hidden layer:

$$Z^{(l+1)} = \gamma g(F) Z^{(l)} S + X \tag{3}$$

where we are given an arbitrary $\gamma \in (0, 1]$ and

$$g(F) = \frac{1}{\|F^\top F\|_{\text{F}} + \epsilon_F} F^\top F \tag{4}$$

with an arbitrary small $\epsilon_F > 0$.

Note that $g(F)$ is constrained by Equation (4) to lie within a Frobenius norm ball of radius $< 1$, which prevents divergence of the infinite sequence. EIGNN extends SGC to an infinite depth model with learnable propagation while additionally adding skip connections. By the designs, EIGNN have several helpful properties: 1) residual connection, which is also used in finite-depth GNN models (e.g., APPNP [15] and GCNII [4]) and has shown its importance in graph learning; 2) learnable weights for propagation on graphs (i.e., g(F)). In contrast, APPNP [15] only directly propagates information without learnable weights.

As EIGNN is a model with infinite depth, how to conduct forward and backward computation for training is not obvious. In the rest of this section, we first show how to perform forward and backward computation for EIGNN. Then we propose to use eigendecomposition to improve the efficiency of this computation.

## 4.1 Forward and backward computation

At first glance, it may seem that we cannot compute the infinite sequence of $(Z^{(l)})_l$ without iterative solvers in practice. However, in Proposition 1, we show that the infinite sequence of $(Z^{(l)})_l$ is guaranteed to converge and is computable without iterative solvers.

**Proposition 1.** *Given any matrix X, F and normalized symmetric adjacency matrix S, the infinite sequence of $(Z^{(l)})_l$ is convergent and the limit of the sequence can be written as follows:*

$$\lim_{H \to \infty} \text{vec}[Z^{(H)}] = (I - \gamma[S \otimes g(F)])^{-1} \text{vec}[X]. \tag{5}$$

This can be proved by using the triangle inequality of a norm and some properties of the Kronecker product. The complete proof can be found in Appendix B.1.

**Computing** $f(X, F, B)$    After getting the limit of the sequence, we can conduct forward computation without iterative solvers as shown in Proposition 2. In contrast, IGNN [10] heavily relies on iterative solvers for both forward and backward computation. The motivation to avoid iterative solvers is that iterative solvers suffer from some commonly known issues, e.g., approximated solutions and sensitive convergence criteria [23].

**Proposition 2.** *Given any matrices X, F and normalized symmetric adjacency matrix S, with an arbitrary $\gamma \in (0, 1]$ and an arbitrary small $\epsilon_F > 0$, $f(X, F, B)$ of Equation (2) can be obtained without iterative solvers:*

$$\text{vec}[f(X, F, B)] = \text{vec}\left[ B \left( \lim_{H \to \infty} Z^{(H)} \right) \right] = [I_n \otimes B] (I - \gamma[S \otimes g(F)])^{-1} \text{vec}[X]. \tag{6}$$

*Proof.* Combining Equation (5) with the property of the vectorization (i.e., $\text{vec}[AB] = (I_m \otimes A) \text{vec}[B]$), Equation (6) can be obtained. □

**Computing** $\frac{\partial(\lim_{H \to \infty} \text{vec}[Z_{X,F}^{(H)}])}{\partial \text{vec}[F]}$    In Proposition 3, we show that backward computation can be done without iterative solvers as well. Without iterative solvers in both forward and backward computation, we can avoid the issue that the solution of iterative methods is not exact. Specifically, as in IGNN [10], the forward computation can yield an error through the iterative solver and the backward pass with the iterative solver can amplify the error from the forward computation. The approximation errors can lead to degradation performance, which we can avoid in our method.

**Proposition 3.** *Given any matrices X, F and normalized symmetric adjacency matrix S, assuming an arbitrary $\gamma \in (0, 1]$ and an arbitrary small $\epsilon_F > 0$, the gradient $\frac{\partial(\lim_{H \to \infty} \text{vec}[Z_{X,F}^{(H)}])}{\partial \text{vec}[F]}$ can be computed without iterative solver:*

$$\frac{\partial(\lim_{H \to \infty} \text{vec}[Z_{X,F}^{(H)}])}{\partial \text{vec}[F]} = \gamma U^{-1} \left[ S \left( \lim_{H \to \infty} Z_{X,F}^{(H)} \right)^\top \otimes I_m \right] \frac{\partial \text{vec}[g(F)]}{\partial \text{vec}[F]}, \tag{7}$$

*where $U = I - \gamma[S \otimes g(F)]$ and the limit can be computed by*

$$\lim_{H \to \infty} \text{vec}[Z_{X,F}^{(H)}] = U^{-1} \text{vec}[X].$$

The proof of Proposition 3 is strongly related to the implicit function theorem. See Appendix B.2 for the complete proof.

**Computing the gradients of objective functions**    With the gradients of the infinite sequence, the gradients of objective functions can be obtained. Define the objective function to be minimized by:

$$L(B, F) = \ell_Y(f(X, F, B)).$$

Here, $\ell_Y$ is an arbitrary differentiable function; for example, it can be set to the cross-entropy loss or square loss with the label matrix $Y \in \mathbb{R}^{m_y \times n}$. Let $\mathbf{f}_{X,F,B} = f(X, F, B)$ and $Z_{X,F} = \lim_{H \to \infty} Z_{X,F}^{(H)}$. Then using the chain rule with Equation (7), we have

$$\frac{\partial L(B, F)}{\partial \operatorname{vec}[F]} = \gamma \frac{\partial \ell_Y(\mathbf{f}_{X,F,B})}{\partial \operatorname{vec}[\mathbf{f}_{X,F,B}]} [I_n \otimes B] U^{-1} \left[ S Z_{X,F}^\top \otimes I_m \right] \frac{\partial \operatorname{vec}[g(F)]}{\partial \operatorname{vec}[F]} \tag{8}$$

and

$$\frac{\partial L(B, F)}{\partial \operatorname{vec}[B]} = \frac{\partial \ell_Y(\mathbf{f}_{X,F,B})}{\partial \operatorname{vec}[\mathbf{f}_{X,F,B}]} [Z_{X,F}^\top \otimes I_{m_y}], \tag{9}$$

where $Z_{X,F} = (I - \gamma[S \otimes g(F)]) \operatorname{vec}[X]$. The full derivation of Equation (8) and (9) can be found in Appendix B.3.

## 4.2    More efficient computation via eigendecomposition

In the previous subsection, we show that even with the infinite depth of EIGNN, forward and backward computations can be conducted without iterative solvers. However, this requires us to calculate and store the matrix $U = I - \gamma[S \otimes g(F)] \in \mathbb{R}^{mn \times mn}$, which can be expensive in terms of memory consumption. For example, when $m = 100, n = 1000$, the memory requirement of $U$ becomes around 40 GB which usually cannot be handled by a commodity GPU. In this subsection, we show that we can achieve more efficient computation via eigendecomposition. We avoid the matrix $U$ by using eigendecomposition of much smaller matrices $g(F) = Q_F \Lambda_F Q_F^\top \in \mathbb{R}^{m \times m}$ and $S = Q_S \Lambda_S Q_S^\top \in \mathbb{R}^{n \times n}$ separately. Therefore, we reduce the memory complexity from an $mn \times mn$ matrix to two matrices with sizes $m \times m$ and $n \times n$ respectively. We first define $G \in \mathbb{R}^{m \times n}$ where $G_{ij} = 1/(1 - \gamma(\bar{\Lambda}_F \bar{\Lambda}_S^\top)_{ij})$. The output of EIGNN and the gradient with respect to $F$ can be computed as follows:

$$f(X, F, B) = B Q_F (G \circ (Q_F^\top X Q_S)) Q_S^\top, \tag{10}$$

$$\nabla_{\operatorname{vec}[F]} L(B, F) = \gamma \left( \frac{\partial \operatorname{vec}[g(F)]}{\partial \operatorname{vec}[F]} \right)^\top \operatorname{vec} \left[ Q_F \left( G \circ \left( Q_F^\top B^\top \frac{\partial \ell_Y(\mathbf{f}_{X,F,B})}{\partial \mathbf{f}_{X,F,B}} Q_S \right) \right) Q_S^\top S Z_{X,F}^\top \right], \tag{11}$$

where $Z_{X,F} = Q_F(G \circ (Q_F^\top X Q_S))Q_S^\top \in \mathbb{R}^{m \times n}$. The complete derivations of Equation (10) and (11) are presented in Appendix B.4.

**Further improvements via avoiding** $\frac{\partial \operatorname{vec}[g(F)]}{\partial \operatorname{vec}[F]}$    As shown in Equation (10) and (11), we can avoid the $mn \times mn$ matrix, but this still requires us to deal with the $mm \times mm$ matrix $\frac{\partial \operatorname{vec}[g(F)]}{\partial \operatorname{vec}[F]}$ in Equation (11). We show that we can further avoid this as well. Therefore, the memory complexity for the computation of the gradient with respect to $F$ reduces from $O(m^4)$ to $O(n^2)$ if $n > m$, or $O(m^2)$ otherwise.

$$\nabla_F L(B, F) = \frac{\gamma}{\|F^\top F\|_{\mathrm{F}} + \epsilon_F} F \left( (R + R^\top) - \frac{2 \langle F^\top F, R \rangle_{\mathrm{F}}}{\|F^\top F\|_{\mathrm{F}}^2 + \epsilon_F \|F^\top F\|_{\mathrm{F}}} F^\top F \right) \in \mathbb{R}^{m \times m}, \tag{12}$$

where

$$R = Q_F \left( G \circ \left( Q_F^\top B^\top \frac{\partial \ell_Y(\mathbf{f}_{X,F,B})}{\partial \mathbf{f}_{X,F,B}} Q_S \right) \right) Q_S^\top S Z_{X,F}^\top \in \mathbb{R}^{m \times m}.$$

Appendix B.5 shows the full derivation of Equation (12). Note that $\frac{\partial \ell_Y(\mathbf{f}_{X,F,B})}{\partial \mathbf{f}_{X,F,B}}$ can be easily calculated by modern autograd frameworks (e.g., PyTorch [20]). For the gradient $\nabla_B L(B, F)$, it can be directly obtained by applying chain rule with $\frac{\partial \ell_Y(\mathbf{f}_{X,F,B})}{\partial \mathbf{f}_{X,F,B}}$ since $B$ is not within the infinite sequence. Thus, we omit the formula here.

### 4.3 Comparison with other GNNs with infinite depth

Comparing with IGNN [10], our model EIGNN can directly obtain a closed-form solution rather than relying on an iterative solver. Thus, we avoid some common issues of iterative solvers [23], e.g., approximated solutions and sensitivity to hyper-parameters. IGNN requires additional conditions to ensure convergence, whereas EIGNN is also an infinite-depth GNN model but avoids such conditions. Without restrictions on the weight matrix, even as a linear model, EIGNN achieves better performance than IGNN as shown in the empirical experiments. Previous work also shows that a linear GNN can achieve comparable performance compared with GCNs [30]. APPNP and PPNP [15] separate predictions from the propagation scheme where PPNP can be treated as considering infinitely many aggregation layers as well via the personalized pagerank matrix. However, the propagation of APPNP and PPNP is not learnable. This is completely different from IGNN and EIGNN, and limits the model's representation power. Besides that, PPNP requires performing a costly matrix inverse operation for a dense $n \times n$ matrix during training.

**Time complexity analysis**    Besides the aforementioned advantages of EIGNN over IGNN, using eigendecomposition also has an efficiency advantage over using iterative solvers (like in IGNN). If we were to use iterative methods in EIGNN, we could iterate Equation (3) until convergence. Each computation of gradients has a theoretical computational complexity of $O(K(m^2 n + mn^2))$ where $K$ is the number of iterations of an iterative method. With eigendecomposition, the time complexity of computing the limit of the infinite sequence is $O(m^3 + m^2 n)$. During training, the eigendecomposition of $g(F)$ requires the cost of $O(m^3)$. Thus, the complexity is $O(m^3 + m^2 n)$ in total. In practice, for large real-world graph settings, the number of nodes $n$ is generally larger than the number of feature dimensions $m$, which makes $m^3 < mn^2$. Therefore, using eigendecomposition for training is more efficient than using iterative methods for infinite-depth GNNs.

In the above analysis, we omit the time complexity of the eigendecomposition of $S$ since it is a one-time preprocessing operation for each graph. Specifically, after obtaining the result of eigendecomposition of $S$, it can always be reused for training and inference. The time complexity of a plain full eigendecomposition of $S$ is $O(n^3)$. In some cases where the complexity of the preprocessing is of importance, we can consider using truncated eigendecomposition or graph coarsening [12] to mitigate the cost. Regarding this, we provide more discussions in Appendix D.

## 5   Experiments

In this section, we demonstrate that EIGNN can effectively learn representations which have the ability to capture long-range dependencies in graphs. Therefore, EIGNN achieves state-of-the-art performance for node classification task on both synthetic and real-world datasets. Specifically, we conduct experiments[1] to compare EIGNN with representative baselines on seven graph datasets (Chains, Chameleon, Squirrel, Cornell, Texas, Wisconsin, and PPI), where Chains is a synthetic dataset used in Gu et al. [10]. Chameleon, Squirrel, Cornell, Texas, and Wisconsin are real-world datasets with a single graph each [21] while PPI is a real-world dataset with multiple graphs [11]. Detailed descriptions of datasets and settings about experiments can be found in Appendix C.

### 5.1   Evaluation on synthetic graphs

**Synthetic experiments & setup**    In order to directly test existing GNN models' abilities for capturing long-range dependencies of graphs, like in Gu et al. [10], we use the Chains dataset, in which the model is supposed to classify nodes on the graph constructed by several chains. The label information is only encoded in the starting end node of each chain. Suppose we have $c$ classes, $n_c$ chains for each class, and $l$ nodes in each chain, then the graph has $c \times n_c \times l$ nodes in total. For training/validation/testing split, we consider 5%/10%/85% which is similar with the semi-supervised node classification setting [14]. We select several representative baselines to compare (i.e., IGNN [10], GCN [14], SGC [30], GAT [26], JKNet [33], APPNP [15], GCNII [4], and H2GCN [36]). The hyper-parameter setting and details about baselines' implementation can be found in Appendix C.2.

---

[1]The implementation can be found at https://github.com/liu-jc/EIGNN

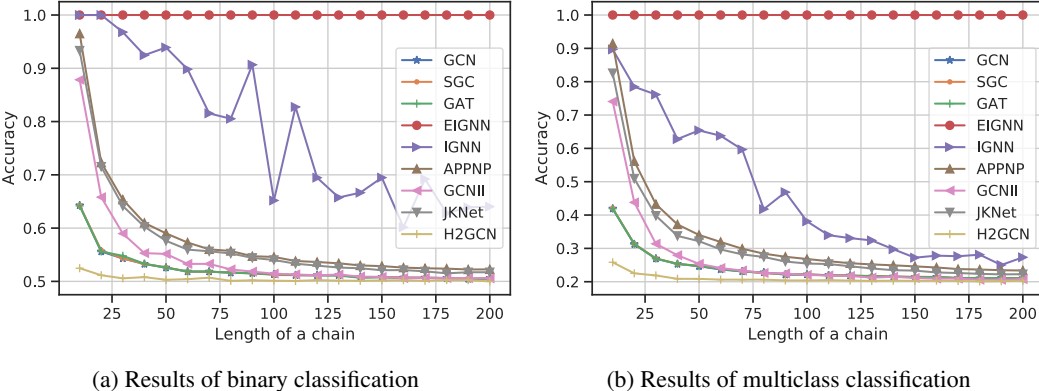

(a) Results of binary classification      (b) Results of multiclass classification

Figure 1: Averaged accuracies with respect to the length of chains.

**Results and analysis** We first consider binary classification ($c = 2$) with 20 chains of each class as in Gu et al. [10]. We conduct experiments on graphs with chains of different lengths (10 to 200 with the interval 10). The averaged accuracies with respect to the length of chains are illustrated in Figure 1a. In general, EIGNN and IGNN always outperform the other baselines. GCN, SGC and GAT provide similarly poor performances, which indicates that these three classic GNN models cannot effectively capture long-range dependencies. H2GCN has an even worse performance than GAT, SGC, GCN, which is caused by its ego- and neighbor-embedding separation focusing more on ego-embedding rather than distant information. APPNP, JKNet, GCNII are either designed to consider different range of neighbors or to mitigate oversmoothing [16, 33]. Despite that they outperform GCN, SGC and GAT, they still perform worse than implicit models with infinitely deep layers (i.e., EIGNN and IGNN) since finite layers for aggregations cannot effectively capture underlying dependencies along extremely long chains (e.g., with length > 30). Intuitively, increasing the number of layers should improve the model's ability to capture long-range dependencies. However, stacking layers requires excessive computational cost and causes oversmoothing which degrades the performance. Appendix C.5 shows the results of APPNP, JKNet and GCNII with a increased number of layers, which illustrates that simply stacking layers cannot effectively help to capture long-range dependencies. Comparing EIGNN and IGNN, they both perfectly capture the long-range dependency when the length is less than 30, while the performance of IGNN generally decreases when the length increases.

In addition to binary classification used in Gu et al. [10], we also conduct experiments on multi-class classification ($c = 5$) to evaluate the ability of capturing long-range dependencies on a generalized setting. Figure 1b shows that EIGNN still easily maintains 100% test accuracy on multi-class classification. In contrast, all other baselines experience performance drops compared to binary classification. On multi-class classification, EIGNN and IGNN consistently outperform the other baselines, verifying that models with infinitely deep layers have advantages for capturing long-range dependencies.

On both classification and multiclass classification settings, as shown in Figure 1, IGNN cannot always achieve 100% test accuracy, especially when the length is longer than 30. However, IGNN is supposed to capture dependencies on infinite-length chains as it is an implicit model with infinite depth. We conjecture that the reason for the inferior performance is that the iterative solver in IGNN cannot easily capture all dependencies on extremely long chains.

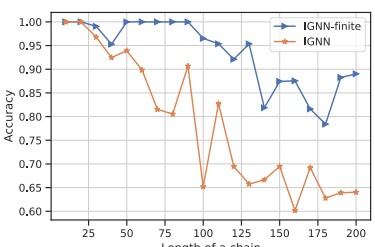

Figure 2: Mean accuracies comparison between the finite-depth and infinite-depth IGNN.

**Finite-depth IGNN v.s. Infinite-depth IGNN** To verify the hypothesis mentioned above, we conduct an additional experiment considering two kinds of IGNN, i.e., finite-depth IGNN and infinite-depth IGNN (the version proposed in Gu et al. [10]). On a dataset with chain length $l$, for finite-depth IGNN, we use $l$ hidden layers which makes it able to capture

Table 2: Results on real-world datasets: mean accuracy (%) ± stdev over different data splits. "*" denotes that the results are obtained from [21].

| | Cornell | Texas | Wisconsin | Chameleon | Squirrel |
|---|---|---|---|---|---|
| **#Nodes** | 183 | 183 | 251 | 2,277 | 5,201 |
| **#Edges** | 280 | 295 | 466 | 31,421 | 198,493 |
| **#Classes** | 5 | 5 | 5 | 5 | 5 |
| EIGNN | **85.13±5.57** | 84.60±5.41 | **86.86±5.54** | **62.92±1.59** | **46.37±1.39** |
| IGNN [10] | 61.35±4.84 | 58.37±5.82 | 53.53±6.49 | 41.38±2.53 | 24.99±2.11 |
| Geom-GCN* [21] | 60.81 | 67.57 | 64.12 | 60.90 | 38.14 |
| SGC [30] | 58.91±3.15 | 58.92±4.32 | 59.41±6.39 | 40.63±2.35 | 28.4±1.43 |
| GCN [14] | 59.19±3.51 | 64.05±5.28 | 61.17±4.71 | 42.34±2.77 | 29.0±1.10 |
| GAT [26] | 59.46±6.94 | 61.62±5.77 | 60.78±8.27 | 46.03±2.51 | 30.51±1.28 |
| APPNP [15] | 63.78±5.43 | 64.32±7.03 | 61.57±3.31 | 43.85±2.43 | 30.67±1.06 |
| JKNet [33] | 58.18±3.87 | 63.78±6.30 | 60.98±2.97 | 44.45±3.17 | 30.83±1.65 |
| GCNII [4] | 76.75±5.95 | 73.51±9.95 | 78.82±5.74 | 48.59±1.88 | 32.20±1.06 |
| H2GCN [36] | 82.22±5.67 | **84.76±5.57** | 85.88±4.58 | 60.30±2.31 | 40.75±1.44 |

dependencies with up to $l$-hops. As shown in Figure 2, IGNN-finite always outperforms IGNN when the length of chains becomes longer. This confirms that the iterative solver used in infinite-depth IGNN is unable to capture all dependencies in longer chains. Note that we use the official implementation[2] of IGNN and follow the exact same setting (e.g., hyperparameters) as in Gu et al. [10]. This can be caused by the approximated solutions generated by iterative solvers and the sensitivity to hyper-parameters of iterative solvers (e.g., convergence criteria). To further support this hypothesis, we illustrate train/test loss trajectories of IGNN-finite and IGNN in Appendix C.3.

**Efficiency comparison** Besides the performance comparison, on Table 1, we also report the training time per epoch of different models (i.e., EIGNN, IGNN, and IGNN-finite) on datasets with varying length $l$ and the number of chains $n_c$. On datasets with different chain length $l$, we set the number of hidden layers to $l$ for IGNN-finite as mentioned in the last paragraph. EIGNN requires less training time than IGNN and IGNN-finite. IGNN-finite with many hidden layers (i.e., >100) spends much more training time than implicit models (i.e., IGNN and EIGNN),

Table 1: Training time per epoch with respect to different $l$ and $n_c$

| $(l, n_c)$ | EIGNN | IGNN | IGNN-finite |
|---|---|---|---|
| (100, 20) | 4.05ms | 27.36ms | 74.61ms |
| (200, 20) | 8.03ms | 29.88ms | 145.78ms |
| (100, 30) | 6.03ms | 64.94ms | 76.31ms |
| (200, 30) | 16.0ms | 60.31ms | 146.60ms |

which shows the efficiency advantage of implicit models when capturing long-range dependencies is desired. Note that EIGNN requires precomputation for eigendecomposition on adjacency matrix. After that, the results can be saved to the disk for further use. Thus, this operation is only conducted once for each dataset, which is tractable and generally requires less than 30 seconds in the experiment.

## 5.2 Evaluation on real-world datasets

**Real-world datasets & setup** Besides the experiments on synthetic graphs, following Pei et al. [21], we also conduct experiments on several real-world heterophilic graphs to evaluate the model's ability to capture long-range dependencies. As nodes with the same class are far away from each other in heterophilic graphs, this requires models to aggregate information from distant nodes. Cornell, Texas, and Wisconsin are web-page graphs of the corresponding universities while Chameleon and Squirrel are web-page graphs of Wikipedia of the corresponding topic.

In addition, to show that EIGNN is applicable to multi-label multi-graph inductive learning, we also evaluate EIGNN on a commonly used dataset Protein-Protein Interation (PPI). PPI contains multiple graphs where nodes are proteins and edges are interactions between proteins. We follow the train/validation/test split used in GraphSAGE [11].

---

[2]https://github.com/SwiftieH/IGNN

**Results and analysis**  As shown in Table 2, in general, EIGNN outperforms other baselines on most datasets. On Texas, EIGNN provides a similar mean accuracy compared with H2GCN which generally provides the second best performance on other real-world datasets. H2GCN focuses more on different designs about aggregations [36]. In contrast, EIGNN performs well through a different approach (i.e., capturing long-range dependencies). In terms of capturing long-range dependencies, H2CN lacks this ability as demonstrated in our synthetic experiments (see Figure 1). Geom-GCN outperforms several baselines (i.e., SGC, GCN, GAT, APPNP and JKNet), which is attributed to its ability to capture long-range dependencies. However, GCNII provides better mean accuracies than Geom-GCN. GCNII with many aggregation layers stacked is designed for mitigating the oversmoothing issue. Thus, this result suggests that alleviating the oversmoothing issue can also improve the model's ability to capture long-range dependencies. Note that Geom-GCN and IGNN are both significantly outperformed by EIGNN. Therefore, it indicates that they are still not effective enough in capturing long-range dependencies on heterophilic graphs.

Table 3 reports the micro-averaged F1 scores of EIGNN and other popular baseline methods on PPI dataset. As we follow the exactly same setting used in Gu et al. [10], the results of baselines are obtained from Gu et al. [10]. From Table 3, EIGNN achieves better performance compared with other baselines, which can be attributed to the ability of EIGNN to capture long-range dependencies between proteins on PPI dataset. Additionally, we also conduct the efficiency comparison between EIGNN and IGNN. On average, for training an epoch, EIGNN requires 2.23 seconds, whereas IGNN spends 35.03 seconds. As suggested in IGNN [10], the training generally requires more than 1000 epochs. Besides the training time, EIGNN needs a one-time preprocessing for eigendecomposition on $S$ which costs 40.59 seconds. Therefore, considering both preprocessing and training, EIGNN still requires less time compared with IGNN.

Table 3: Multi-label node classification on PPI: Micro-F1 (%).

| Method | Micro-F1 |
|---|---|
| Multi-Layer Perceptron | 46.2 |
| GCN [14] | 59.2 |
| GraphSAGE [11] | 78.6 |
| SSE [6] | 83.6 |
| GAT [26] | 97.3 |
| IGNN [10] | 97.6 |
| **EIGNN** | **98.0** |

## 5.3  Noise sensitivity

Recent research [31, 37] shows that GNNs are vulnerable to perturbations, and El Ghaoui et al. [7] analyze the potential robustness properties of an implicit model. Thus, in this subsection, we empirically examine the potential benefit of EIGNN on robustness against noise. For simplicity, we add perturbations only to node features. Experiments on synthetic datasets and real-world datasets are both conducted.

**Synthetic experiments**  On synthetic chain datasets, we add random perturbations to node features as in Wu et al. [31]. Specifically, we add uniform noise $\epsilon \sim \mathcal{U}(-\alpha, \alpha)$ to each node's features for constructing noisy datasets. Figure 3 shows that across different noise levels, EIGNN consistently outperforms IGNN. This indicates that EIGNN is less vulnerable to noisy features.

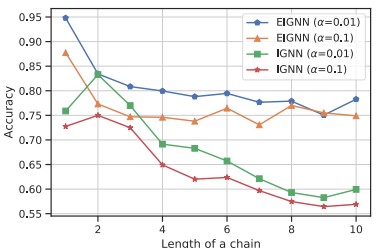

Figure 3: Accuracy on datasets with different feature noise.

**Real-world experiments**  On real-world datasets, we use two classic adversarial attacks (i.e., FGSM [8] and PGD [18]) to add perturbations on node features. See the detailed setting in Appendix C.2. We compare three models (i.e., EIGNN, H2GCN, IGNN) on Cornell, Texas, and Wisconsin under different level perturbations. As we can observe from Table 4, EIGNN consistently achieves the best performance under different perturbation rates on all three datasets. Specifically, EIGNN outperforms H2GCN by a larger margin when the perturbation rate is higher. Under FGSM with a perturbation rate larger than 0.001, H2GCN is even outperformed by IGNN which has worse performance under 0.0001 perturbation rate.

In summary, the above results show that EIGNN is generally more robust against perturbations compared with H2GCN. The intuitive reason is that EIGNN is not/less sensitive to local neighborhoods,

Table 4: Accuracy (%) under attacks with different perturbations.

| | Attack | FGSM | | | PGD | | |
|---|---|---|---|---|---|---|---|
| | Perturbation | 0.0001 | 0.001 | 0.01 | 0.0001 | 0.001 | 0.01 |
| Corn. | EIGNN | **85.13±5.57** | **84.05±5.59** | **72.70±6.56** | **85.13±5.57** | **84.05±5.59** | **73.24±6.45** |
| | H2GCN | 79.46±5.16 | 26.49±8.62 | 2.16±2.36 | 82.97±5.27 | 65.13±9.00 | 25.68±8.65 |
| | IGNN | 61.08±3.66 | 60.54±4.22 | 57.30±5.51 | 61.07±3.66 | 60.54±4.22 | 57.84±5.16 |
| Texas | EIGNN | **84.33±5.77** | **83.79±5.26** | **74.59±4.05** | 84.33±5.77 | **83.79±5.27** | **75.14±4.32** |
| | H2GCN | 81.35±6.22 | 39.79±9.05 | 7.30±7.74 | **84.86±5.57** | 70.54±6.89 | 36.22±7.57 |
| | IGNN | 57.84±5.16 | 57.28±5.64 | 53.51±5.10 | 57.84±5.16 | 57.30±5.64 | 53.51±5.10 |
| Wisc. | EIGNN | **86.67±6.19** | **84.90±6.45** | **70.00±6.73** | **86.67±6.19** | **85.10±6.34** | **71.96±6.76** |
| | H2GCN | 83.53±4.04 | 47.84±6.51 | 4.12±3.66 | 84.90±3.62 | 77.25±4.13 | 38.04±6.52 |
| | IGNN | 53.53±4.39 | 53.14±4.06 | 47.26±3.77 | 53.53±4.39 | 53.14±4.06 | 47.84±3.53 |

while finite-depth GNNs (e.g., H2GCN) could be very sensitive to local neighborhoods. Finite-depth GNNs usually only have a few layers and the node representations would be changed a lot when aggregating from perturbed node features. In contrast, EIGNN, as an infinite-depth GNN model, can consider more about global information and become less sensitive to perturbations from local neighborhoods.

## 6 Conclusion

In this paper, we address an important limitation of existing GNNs: their lack of ability to capture long-range dependencies. We propose EIGNN, a GNN model with implicit infinite layers. Instead of using iterative solvers like in previous work, we derive a closed-form solution of EIGNN with rigorous proofs, which makes training tractable. We further achieve more efficient computation for training our model by using eigendecomposition. On synthetic experiments, we demonstrate that EIGNN has a better ability to capture long-range dependencies. In addition, it also provides state-of-the-art performance on real-world datasets. Moreover, our model is less susceptible to perturbations on node features compared with other models. One of the potential limitations of our work is that EIGNN can be slow when the number of feature dimensions is very large (e.g., 1 million), which is rare in practical GNN settings. Future work could propose a new model which allows for efficient training even with large feature dimension.

## Acknowledgments and Disclosure of Funding

We would like to thank Keke Huang and Tianyuan Jin for helpful discussions. We also gratefully acknowledge the insightful feedback and suggestions from the anonymous reviewers. This paper is supported by the Ministry of Education, Singapore (Grant No: MOE2018-T2-2-091). The views and conclusions contained in this paper are those of the authors and should not be interpreted as representing any funding agencies.

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
