# A   Kronecker Product and Vectorization

Given two matrices $A \in \mathbb{R}^{m \times n}$ and $B \in \mathbb{R}^{p \times q}$, the Kronecker product $A \otimes B \in \mathbb{R}^{pm \times qn}$ is defined as follows:

$$A \otimes B = \begin{pmatrix} A_{11}B & \cdots & A_{1n}B \\ \vdots & \ddots & \vdots \\ A_{m1}B & \cdots & A_{mn}B \end{pmatrix}.$$

One of the most important properties of the vectorization with the kronecker product is $\mathrm{vec}[AB] = (I_m \otimes A)\,\mathrm{vec}[B]$ where the vectorization of $B$ (i.e., $\mathrm{vec}[B] \in \mathbb{R}^{mn}$) can be obtained by stacking the columns of the matrix $B \in \mathbb{R}^{m \times n}$ into a single column vector. We summarize a few other properties of kronecker product and vectorization used in our proofs as follows:

- $\|A \otimes B\|_2 = \|A\|_2 \|B\|_2$
- $[A \otimes B]^\top = A^\top \otimes B^\top$
- $\mathrm{vec}[ABC] == (C^\top \otimes A)\,\mathrm{vec}[B] = (I_n \otimes AB)\,\mathrm{vec}[C] = (C^\top B^\top \otimes I_k)\,\mathrm{vec}[A]$
- $[AB] \otimes [CD] = [A \otimes C][B \otimes D]$

# B   Proofs

## B.1   Proof of Proposition 1

*Proof.* Since $S$ is symmetric,

$$\mathrm{vec}[Z^{(l+1)}] = \mathrm{vec}[X] + \gamma\,\mathrm{vec}[g(F)Z^{(l)}S] = \gamma[S \otimes g(F)]\,\mathrm{vec}[Z^{(l)}] + \mathrm{vec}[X].$$

By repeatedly apply this to $Z^{(l)}$,

$$\mathrm{vec}[Z^{(H)}] = \sum_{k=0}^{H} \gamma^k [S \otimes g(F)]^k\,\mathrm{vec}[X] = \sum_{k=0}^{H} \gamma^k [S^k \otimes g(F)^k]\,\mathrm{vec}[X].$$

Let $s_H = \sum_{k=0}^{H} \gamma^k [S^k \otimes g(F)^k]$. Then for $H > l$, by using the triangle inequality of a norm and the submultiplicativity of an induced matrix norm,

$$\begin{aligned}
\|s_H - s_l\|_2 = \left\| \sum_{k=l+1}^{H} \gamma^k [S^k \otimes g(F)^k] \right\|_2 &\leq \sum_{k=l+1}^{H} \left\| \gamma^k [S^k \otimes g(F)^k] \right\|_2 \\
&\leq \sum_{k=l+1}^{H} \prod_{i=1}^{k} \left\| \gamma[S \otimes g(F)] \right\|_2 \\
&= \sum_{k=l+1}^{H} \prod_{i=1}^{k} \gamma \|S\|_2 \|g(F)\|_{\mathrm{F}}
\end{aligned}$$

where the last line follows from the property of the Kronecker product. Since $S$ is symmetric and normalized,

$$\|S\|_2 = |\lambda_{\max}(S)| \leq 1.$$

Since $\epsilon_F > 0$,

$$\|g(F)\|_2 \leq \|g(F)\|_{\mathrm{F}} = \left\| \frac{1}{\|F^\top F\|_{\mathrm{F}} + \epsilon_F} F^\top F \right\|_{\mathrm{F}} = \frac{\|F^\top F\|_{\mathrm{F}}}{\|F^\top F\|_{\mathrm{F}} + \epsilon_F} < 1.$$

In other words, there exists $\delta \in [0, 1)$ such that $\|g(F)\|_2 \leq \delta$. Combining,

$$\|s_H - s_l\|_2 \leq \sum_{k=l+1}^{H} \prod_{i=1}^{k} \gamma \|S\|_2 \|g(F)\|_{\mathrm{F}} \leq \sum_{k=l+1}^{H} \prod_{i=1}^{k} \gamma\delta = \sum_{k=l+1}^{H} (\gamma\delta)^k.$$

Here, we have $\gamma\delta \in [0,1)$ since $\gamma \in (0,1]$ and $\delta \in [0,1)$. Thus, we have $\sum_{k=0}^{H}(\gamma\delta)^k = \frac{1-(\gamma\delta)^{H+1}}{1-\gamma\delta}$. Therefore,

$$\sum_{k=l+1}^{H}(\gamma\delta)^k = \sum_{k=0}^{H}(\gamma\delta)^k - \sum_{k=0}^{l}(\gamma\delta)^k = \frac{(\gamma\delta)^{l+1} - (\gamma\delta)^{H+1}}{1-\gamma\delta} \leq \frac{(\gamma\delta)^{l+1}}{1-\gamma\delta}.$$

Combining

$$\|s_H - s_l\|_2 \leq \sum_{k=l+1}^{H}(\gamma\delta)^k \leq \frac{(\gamma\delta)^{l+1}}{1-\gamma\delta}.$$

Since $\|s_H - s_l\|_2 \leq \frac{(\gamma\delta)^{l+1}}{1-\gamma\delta}$ with $\gamma\delta < 1$, we have that for any $\epsilon > 0$, there exists an integer $\bar{l}$ such that $\|s_H - s_l\|_2 < \epsilon$ for any $H, l \geq \bar{l}$. Thus, the sequence $(s_l)_l = (\sum_{k=0}^{l}\gamma^k[S^k \otimes g(F)^k])_l$ is a Cauchy sequence and converges.

Since it converges, by recalling the fact that for an matrix $M$,

$$(1 - M)\left(\sum_{k=0}^{H}M^k\right) = I - M^{H+1},$$

we have that

$$(I - \gamma[S \otimes g(F)])\left(\lim_{H\to\infty}\sum_{k=0}^{H}\gamma^k[S^k \otimes g(F)^k]\right) = \lim_{H\to\infty}(I - \gamma[S \otimes g(F)])\left(\sum_{k=0}^{H}\gamma^k[S^k \otimes g(F)^k]\right)$$
$$= \lim_{H\to\infty}(I - \gamma^{H+1}[S^{H+1} \otimes g(F)^{H+1}])$$
$$= I$$

where the last line follows from the fact that

$$\|\gamma^{H+1}[S^{H+1}\otimes g(F)^{H+1}]\|_2 \leq \prod_{i=1}^{H}\|\gamma[S \otimes g(F)]\|_2 = \prod_{i=1}^{H}\gamma\|S\|_2\|g(F)\|_{\mathrm{F}} \leq (\gamma\delta)^H \to 0 \quad \text{as } H \to \infty$$

where the inequalities follows from the above equations used to bound $\|s_H - s_l\|_2$.

This implies that

$$\left(\lim_{H\to\infty}\sum_{k=0}^{H}\gamma^k[S^k \otimes g(F)^k]\right) = (I - \gamma[S \otimes g(F)])^{-1},$$

if the inverse $(I - \gamma[S \otimes g(F)])^{-1}$ exists. This inverse exists, which can be seen by the following facts. From the above equations used to bound $\|s_H - s_l\|_2$ (and the fact that $S$ and $g(F)$ are symmetric and hence their singular values are the absolute values of their eigenvalues), we have $\lambda_{\max}(\gamma[S \otimes g(F)]) = \gamma\lambda_{\max}(S)\lambda_{\max}(g(F)) < 1$. Also, since $(I - \gamma[S \otimes g(F)])$ is real symmetric (as $[S \otimes g(F)]^\top = [S^\top \otimes g(F)^\top] = [S \otimes g(F)]$), all the eigenvalues of the matrix $(I - \gamma[S \otimes g(F)])$ are real numbers. Combining these facts, all the eigenvalues of $(I - \gamma[S \otimes g(F)])$ are strictly positive (since the eigenvalues of the identity matrix $I$ are ones). Therefore, $(I - \gamma[S \otimes g(F)])$ is positive definite and invertible.

Using these results, we have

$$\lim_{H\to\infty}\mathrm{vec}[Z^{(H)}] = (I - \gamma[S \otimes g(F)])^{-1}\mathrm{vec}[X].$$

$$\mathrm{vec}[f(X, F, B)] = \mathrm{vec}\left[B\left(\lim_{H\to\infty}Z^{(H)}\right)\right] = [I_n \otimes B]\,\mathrm{vec}\left[\left(\lim_{H\to\infty}Z^{(H)}\right)\right] = [I_n \otimes B]\,(I - \gamma[S \otimes g(F)])^{-1}\mathrm{vec}[X].$$

$\square$

## B.2 Proof of Proposition 3

*Proof.* By the definition of $Z^{(H)} = Z^{(H)}_{X,F}$,

$$\lim_{H\to\infty} \text{vec}[Z^{(H)}] = \lim_{H\to\infty} \text{vec}[\gamma g(F)Z^{(H-1)}S + X] = \gamma[S \otimes g(F)]\left(\lim_{H\to\infty} \text{vec}[Z^{(H-1)}]\right) + \text{vec}[X].$$

Since we have shown that the limit exists in Proposition 1, we can define $Z_{X,F} = \lim_{H\to\infty} Z^{(H)}$. Then the above equation can be rewritten as

$$\text{vec}[Z_{X,F}] = \gamma[S \otimes g(F)]\,\text{vec}[Z_{X,F}] + \text{vec}[X],$$

which is equivalent to

$$\text{vec}[Z_{X,F}] - \gamma[S \otimes g(F)]\,\text{vec}[Z_{X,F}] - \text{vec}[X] = 0.$$

Define a function

$$\varphi(\text{vec}[F], \text{vec}[Z]) = \text{vec}[Z] - \gamma[S \otimes g(F)]\,\text{vec}[Z] - \text{vec}[X],$$

where $F$ and $Z$ are independent variables of this function $\varphi$. For general $F$ and $Z$, $\varphi(\text{vec}[F], \text{vec}[Z])$ is allowed to be nonzero. In contrast, we have $\varphi(\text{vec}[F], \text{vec}[Z_{X,F}]) = 0$ where $F$ and $Z_{X,F}$ are dependent variables because of the definition of $Z_{X,F} = \lim_{H\to\infty} Z^{(H)}_{X,F}$.

With this definition, we have

$$J(\bar{F}, \bar{Z}) = \left.\frac{\partial \varphi(\text{vec}[F], \text{vec}[Z])}{\partial \text{vec}[Z]}\right|_{(F,Z)=(\bar{F},\bar{Z})} = I - \gamma[S \otimes g(\bar{F})].$$

Here, $\varphi(\text{vec}[F], \text{vec}[Z_{X,F}]) = 0$ and the Jacobian $J(F, Z_{X,F}) = I - \gamma[S \otimes g(F)]$ is invertible as shown in Appendix B.1. Therefore, applying the implicit function theorem to the function $\varphi$ yields the following: there exists an open set containing $F$ and a function $\psi$ defined on the open set such that $\psi(F) = \text{vec}[Z_{X,F}]$, $\varphi(\text{vec}[\bar{F}], \psi(\bar{F})) = 0$ for all $\bar{F}$ in the open set, and $\frac{\partial \psi(\bar{F})}{\partial \text{vec}[\bar{F}]} = -J(\bar{F}, \bar{Z})^{-1}\frac{\partial \varphi(\text{vec}[\bar{F}], \text{vec}[\bar{Z}])}{\partial \text{vec}[\bar{F}]}$ for all $\bar{F}$ in the open set. From the above equivalent definition of $Z_{X,F}$ (i.e., $\varphi(\text{vec}[F], \text{vec}[Z_{X,F}]) = 0$), this implies that

$$\frac{\partial \text{vec}[Z_{X,F}]}{\partial \text{vec}[F]} = \left.-J(F, Z_{X,F})^{-1}\frac{\partial \varphi(\text{vec}[F], \text{vec}[Z])}{\partial \text{vec}[F]}\right|_{Z=Z_{X,F}}$$

Using the definition of $\varphi$

$$\begin{aligned}
\frac{\partial \varphi(\text{vec}[F], \text{vec}[Z])}{\partial \text{vec}[F]} &= \frac{\partial}{\partial \text{vec}[F]}\,\text{vec}[Z] - \gamma\,\text{vec}[g(F)ZS] - \text{vec}[X] \\
&= \frac{\partial}{\partial \text{vec}[F]}\,\text{vec}[Z] - \gamma[SZ^\top \otimes I_m]\,\text{vec}[g(F)] - \text{vec}[X] \\
&= -\gamma[SZ^\top \otimes I_m]\frac{\partial \text{vec}[g(F)]}{\partial \text{vec}[F]}
\end{aligned}$$

Combining,

$$\frac{\partial \text{vec}[Z_{X,F}]}{\partial \text{vec}[F]} = \left.-J(F, Z_{X,F})^{-1}\frac{\partial \varphi(\text{vec}[F], \text{vec}[Z])}{\partial \text{vec}[F]}\right|_{Z=Z_{X,F}} = \gamma(I - \gamma[S \otimes g(F)])^{-1}[SZ^\top_{X,F} \otimes I_m]\frac{\partial \text{vec}[g(F)]}{\partial \text{vec}[F]},$$

where $Z_{X,F} = \lim_{H\to\infty} Z^{(H)}_{X,F}$ and Proposition 1 has shown that

$$\lim_{H\to\infty} \text{vec}[Z^{(H)}_{X,F}] = (I - \gamma[S \otimes g(F)])^{-1}\,\text{vec}[X].$$

$\square$

## B.3 Derivations of Equation (8) and (9)

By the chain rule,

$$\frac{\partial L(B,F)}{\partial \operatorname{vec}[F]} = \frac{\partial \ell_Y(\mathbf{f}_{X,F,B})}{\partial \operatorname{vec}[\mathbf{f}_{X,F,B}]} \frac{\partial \operatorname{vec}[\mathbf{f}_{X,F,B}]}{\partial \operatorname{vec}[F]}.$$

Since $\operatorname{vec}[\mathbf{f}_{X,F,B}] = [I_n \otimes B] \operatorname{vec}[Z_{X,F}]$,

$$\frac{\partial L(B,F)}{\partial \operatorname{vec}[F]} = \frac{\partial \ell_Y(\mathbf{f}_{X,F,B})}{\partial \operatorname{vec}[\mathbf{f}_{X,F,B}]} \frac{\partial \operatorname{vec}[\mathbf{f}_{X,F,B}]}{\partial \operatorname{vec}[F]} = \frac{\partial \ell_Y(\mathbf{f}_{X,F,B})}{\partial \operatorname{vec}[\mathbf{f}_{X,F,B}]} [I_n \otimes B] \frac{\partial \operatorname{vec}[Z_{X,F}]}{\partial \operatorname{vec}[F]}.$$

Using the formula of $\frac{\partial \operatorname{vec}[Z_{X,F}]}{\partial \operatorname{vec}[F]}$ from Appendix B.2,

$$\frac{\partial L(B,F)}{\partial \operatorname{vec}[F]} = \gamma \frac{\partial \ell_Y(\mathbf{f}_{X,F,B})}{\partial \operatorname{vec}[\mathbf{f}_{X,F,B}]} [I_n \otimes B] U^{-1} \left[ S Z_{X,F}^\top \otimes I_m \right] \frac{\partial \operatorname{vec}[g(F)]}{\partial \operatorname{vec}[F]}.$$

Similarly, since $\operatorname{vec}[\mathbf{f}_{X,F,B}] = [Z_{X,F}^\top \otimes I_{m_y}] \operatorname{vec}[B]$,

$$\frac{\partial L(B,F)}{\partial \operatorname{vec}[B]} = \frac{\partial \ell_Y(\mathbf{f}_{X,F,B})}{\partial \operatorname{vec}[\mathbf{f}_{X,F,B}]} \frac{\partial \operatorname{vec}[\mathbf{f}_{X,F,B}]}{\partial \operatorname{vec}[B]} = \frac{\partial \ell_Y(\mathbf{f}_{X,F,B})}{\partial \operatorname{vec}[\mathbf{f}_{X,F,B}]} [Z_{X,F}^\top \otimes I_{m_y}]$$

## B.4 Derivations of Equation (10) and (11)

The matrix $g(F)$ is real symmetric since

$$g(F)^\top = \left( \frac{1}{\|F^\top F\|_{\mathrm{F}} + \epsilon_F} F^\top F \right)^\top = \frac{1}{\|F^\top F\|_{\mathrm{F}} + \epsilon_F} F^\top F = g(F).$$

Since $S$ is also real symmetric, we can use eigendecomposition of $g(F)$ and $S$ as

$$g(F) = Q_F \Lambda_F Q_F^\top$$
$$S = Q_S \Lambda_S Q_S^\top$$

where $Q_S$ and $Q_F$ are unitary matrices. Thus,

$$[S \otimes g(F)] = [Q_S \Lambda_S Q_S^\top \otimes Q_F \Lambda_F Q_F^\top] = [Q_S \otimes Q_F][\Lambda_S \otimes \Lambda_F][Q_S^\top \otimes Q_F^\top].$$
$$I_{mn} = [I_n \otimes I_m] = [Q_S Q_S^\top \otimes Q_F Q_F^\top] = [Q_S \otimes Q_F][I_n \otimes I_m][Q_S^\top \otimes Q_F^\top].$$

Combining,

$$\begin{aligned}
U &= I_{mn} - \gamma[S \otimes g(F)] \\
&= [Q_S \otimes Q_F][I_n \otimes I_m][Q_S^\top \otimes Q_F^\top] - \gamma[Q_S \otimes Q_F][\Lambda_S \otimes \Lambda_F][Q_S^\top \otimes Q_F^\top] \\
&= [Q_S \otimes Q_F]([I_n \otimes I_m] - \gamma[\Lambda_S \otimes \Lambda_F])[Q_S^\top \otimes Q_F^\top].
\end{aligned}$$

Thus,

$$\begin{aligned}
U^{-1} &= [Q_S \otimes Q_F]([I_n \otimes I_m] - \gamma[\Lambda_S \otimes \Lambda_F])^{-1}[Q_S^\top \otimes Q_F^\top] \\
&= [Q_S \otimes Q_F] \operatorname{diag}(\operatorname{vec}[G])[Q_S^\top \otimes Q_F^\top],
\end{aligned}$$

where the matrix $G \in \mathbb{R}^{m \times n}$ is defined by $G_{ij} = [([I_n \otimes I_m] - \gamma[\Lambda_S \otimes \Lambda_F])^{-1}]_{(i+(j-1)m)(i+(j-1)m)}$ (so that $\operatorname{vec}[G]_i = [([I_n \otimes I_m] - \gamma[\Lambda_S \otimes \Lambda_F])^{-1}]_{ii}$ for $i = 1, 2, \ldots, mn$). By the definition of Kronecker product, this is equivalent to $G_{ij} = 1/(1 - \gamma(\bar{\Lambda}_F \bar{\Lambda}_S^\top)_{ij})$.

Using this form of $U^{-1}$,

$$\begin{aligned}
\operatorname{vec}[f(X,F,B)] &= \operatorname{vec}\left[ B \left( \lim_{H \to \infty} Z^{(H)} \right) \right] \\
&= [I_n \otimes B] U^{-1} \operatorname{vec}[X] \\
&= [I_n \otimes B][Q_S \otimes Q_F] \operatorname{diag}(\operatorname{vec}[G])[Q_S^\top \otimes Q_F^\top] \operatorname{vec}[X] \\
&= [Q_S \otimes B Q_F] \operatorname{diag}(\operatorname{vec}[G]) \operatorname{vec}[Q_F^\top X Q_S] \\
&= [Q_S \otimes B Q_F](\operatorname{vec}[G] \circ \operatorname{vec}[Q_F^\top X Q_S]) \\
&= [Q_S \otimes B Q_F] \operatorname{vec}[G \circ (Q_F^\top X Q_S)] \\
&= \operatorname{vec}[B Q_F (G \circ (Q_F^\top X Q_S)) Q_S^\top] \in \mathbb{R}^{m_y n}
\end{aligned}$$

Therefore,

$$\mathbf{f}_{X,F,B} = f(X,F,B) = BQ_F(G \circ (Q_F^\top X Q_S))Q_S^\top \in \mathbb{R}^{m_y \times n}.$$

Similarly,

$$\mathrm{vec}[Z_{X,F}] = \left( \lim_{H \to \infty} Z^{(H)} \right) = U^{-1} \mathrm{vec}[X] = [Q_S \otimes Q_F] \, \mathrm{diag}(\mathrm{vec}[G])[Q_S^\top \otimes Q_F^\top] \, \mathrm{vec}[X]$$

$$= [Q_S \otimes Q_F] \, \mathrm{vec}[G \circ (Q_F^\top X Q_S)]$$

$$= \mathrm{vec}[Q_F(G \circ (Q_F^\top X Q_S))Q_S^\top] \in \mathbb{R}^{mn}$$

Therefore,

$$Z_{X,F} = Q_F(G \circ (Q_F^\top X Q_S))Q_S^\top \in \mathbb{R}^{m \times n}.$$

Moreover, using the form of $U^{-1}$,

$$\frac{\partial L(B,F)}{\partial \, \mathrm{vec}[F]} = \gamma \frac{\partial \ell_Y(\mathbf{f}_{X,F,B})}{\partial \, \mathrm{vec}[\mathbf{f}_{X,F,B}]} [I_n \otimes B] U^{-1} \left[ SZ_{X,F}^\top \otimes I_m \right] \frac{\partial \, \mathrm{vec}[g(F)]}{\partial \, \mathrm{vec}[F]}$$

$$= \gamma \frac{\partial \ell_Y(\mathbf{f}_{X,F,B})}{\partial \, \mathrm{vec}[\mathbf{f}_{X,F,B}]} [I_n \otimes B][Q_S \otimes Q_F] \, \mathrm{diag}(\mathrm{vec}[G])[Q_S^\top \otimes Q_F^\top] \left[ SZ_{X,F}^\top \otimes I_m \right] \frac{\partial \, \mathrm{vec}[g(F)]}{\partial \, \mathrm{vec}[F]}$$

$$= \gamma \frac{\partial \ell_Y(\mathbf{f}_{X,F,B})}{\partial \, \mathrm{vec}[\mathbf{f}_{X,F,B}]} [Q_S \otimes BQ_F] \, \mathrm{diag}(\mathrm{vec}[G]) \left[ Q_S^\top SZ_{X,F}^\top \otimes Q_F^\top \right] \frac{\partial \, \mathrm{vec}[g(F)]}{\partial \, \mathrm{vec}[F]}.$$

Thus,

$$\nabla_{\mathrm{vec}[F]} L(B,F) = \left( \frac{\partial L(B,F)}{\partial \, \mathrm{vec}[F]} \right)^\top$$

$$= \gamma \left( \frac{\partial \, \mathrm{vec}[g(F)]}{\partial \, \mathrm{vec}[F]} \right)^\top \left[ Q_S^\top SZ_{X,F}^\top \otimes Q_F^\top \right]^\top \mathrm{diag}(\mathrm{vec}[G]) \left[ Q_S \otimes BQ_F \right]^\top \left( \frac{\partial \ell_Y(\mathbf{f}_{X,F,B})}{\partial \, \mathrm{vec}[\mathbf{f}_{X,F,B}]} \right)^\top$$

$$= \gamma \left( \frac{\partial \, \mathrm{vec}[g(F)]}{\partial \, \mathrm{vec}[F]} \right)^\top [Z_{X,F} S Q_S \otimes Q_F] \, \mathrm{diag}(\mathrm{vec}[G]) [ Q_S^\top \otimes Q_F^\top B^\top] \, \mathrm{vec} \left[ \frac{\partial \ell_Y(\mathbf{f}_{X,F,B})}{\partial \mathbf{f}_{X,F,B}} \right]$$

$$= \gamma \left( \frac{\partial \, \mathrm{vec}[g(F)]}{\partial \, \mathrm{vec}[F]} \right)^\top [Z_{X,F} S Q_S \otimes Q_F] \, \mathrm{diag}(\mathrm{vec}[G]) \, \mathrm{vec} \left[ Q_F^\top B^\top \frac{\partial \ell_Y(\mathbf{f}_{X,F,B})}{\partial \mathbf{f}_{X,F,B}} Q_S \right]$$

$$= \gamma \left( \frac{\partial \, \mathrm{vec}[g(F)]}{\partial \, \mathrm{vec}[F]} \right)^\top [Z_{X,F} S Q_S \otimes Q_F] \, \mathrm{vec} \left[ G \circ \left( Q_F^\top B^\top \frac{\partial \ell_Y(\mathbf{f}_{X,F,B})}{\partial \mathbf{f}_{X,F,B}} Q_S \right) \right]$$

$$= \gamma \left( \frac{\partial \, \mathrm{vec}[g(F)]}{\partial \, \mathrm{vec}[F]} \right)^\top \mathrm{vec} \left[ Q_F \left( G \circ \left( Q_F^\top B^\top \frac{\partial \ell_Y(\mathbf{f}_{X,F,B})}{\partial \mathbf{f}_{X,F,B}} Q_S \right) \right) Q_S^\top SZ_{X,F}^\top \right] \in \mathbb{R}^{mm}$$

### B.5 Derivation of Equation (12)

By using chain rule,

$$\frac{\partial \, \mathrm{vec}[g(F)]}{\partial \, \mathrm{vec}[F]} = \frac{\partial}{\partial \, \mathrm{vec}[F]} \frac{1}{\|F^\top F\|_\mathrm{F} + \epsilon_F} \mathrm{vec}[F^\top F]$$

$$= \frac{\partial}{\partial \, \mathrm{vec}[F]} \frac{1}{\sqrt{\mathrm{vec}[F^\top F]^\top \, \mathrm{vec}[F^\top F]} + \epsilon_F} \mathrm{vec}[F^\top F]$$

$$= \left( \frac{\partial \frac{1}{\sqrt{v^\top v} + \epsilon_F} v}{\partial v} \Bigg|_{v = \mathrm{vec}[F^\top F]} \right) \frac{\partial \, \mathrm{vec}[F^\top F]}{\partial \, \mathrm{vec}[F]}.$$

For the first term, using chain rule,

$$
\frac{\partial \frac{1}{\sqrt{v^\top v}+\epsilon_F} v}{\partial v} = \left( \frac{\partial av}{\partial a} \bigg|_{a=\frac{1}{\sqrt{v^\top v}+\epsilon_F}} \right) \left( \frac{\partial \frac{1}{\sqrt{v^\top v}+\epsilon_F}}{\partial v} \right) + \frac{\partial av}{\partial v} \bigg|_{a=\frac{1}{\sqrt{v^\top v}+\epsilon_F}}
$$

$$
= v \left( \frac{\partial \frac{1}{\sqrt{v^\top v}+\epsilon_F}}{\partial v} \right) + \frac{1}{\sqrt{v^\top v}+\epsilon_F} I_{mm}
$$

$$
= v \left( \frac{\partial a^{-1}}{\partial a} \bigg|_{a=\sqrt{v^\top v}+\epsilon_F} \right) \left( \frac{\partial \sqrt{v^\top v}+\epsilon_F}{\partial v} \right) + \frac{1}{\sqrt{v^\top v}+\epsilon_F} I_{mm}
$$

$$
= v \left( -(\sqrt{v^\top v}+\epsilon_F)^{-2} \right) \left( \frac{\partial \sqrt{v^\top v}}{\partial v} \right) + \frac{1}{\sqrt{v^\top v}+\epsilon_F} I_{mm}
$$

$$
= v \left( -(\sqrt{v^\top v}+\epsilon_F)^{-2} \right) \left( \frac{\partial \sqrt{a}}{\partial a} \bigg|_{a=v^\top v} \right) \left( \frac{\partial v^\top v}{\partial v} \right) + \frac{1}{\sqrt{v^\top v}+\epsilon_F} I_{mm}
$$

$$
= v \left( -(\sqrt{v^\top v}+\epsilon_F)^{-2} \right) \left( \frac{1}{2}(v^\top v)^{-1/2} \right) \left( 2v^\top \right) + \frac{1}{\sqrt{v^\top v}+\epsilon_F} I_{mm}
$$

$$
= -\frac{1}{(\sqrt{v^\top v}+\epsilon_F)^2} \frac{1}{\sqrt{v^\top v}} vv^\top + \frac{1}{\sqrt{v^\top v}+\epsilon_F} I_{mm}
$$

$$
= \frac{1}{\sqrt{v^\top v}+\epsilon_F} \left( I_{mm} - \frac{1}{(\sqrt{v^\top v}+\epsilon_F)\sqrt{v^\top v}} vv^\top \right)
$$

Thus,

$$
\frac{\partial \frac{1}{\sqrt{v^\top v}+\epsilon_F} v}{\partial v} \bigg|_{v=\mathrm{vec}[F^\top F]} = \frac{1}{\|F^\top F\|_F+\epsilon_F} \left( I_{mm} - \frac{1}{(\|F^\top F\|_F+\epsilon_F)\|F^\top F\|_F} \mathrm{vec}[F^\top F] \mathrm{vec}[F^\top F]^\top \right)
$$

For the second term,

$$
\frac{\partial F^\top F}{\partial F_{ij}} = F^\top \Delta^{ij} + \Delta^{ji} F,
$$

where $\Delta^{ij} \in \mathbb{R}^{m \times m}$ is the matrix with the $(i,j)$-th entry being one and all other entries being zero. Using vectorization and the square commutation matrix $K^{(m,m)}$,

$$
\frac{\partial \mathrm{vec}[F^\top F]}{\partial F_{ij}} = \mathrm{vec}[F^\top \Delta^{ij}] + \mathrm{vec}[\Delta^{ji} F]
$$

$$
= \mathrm{vec}[F^\top \Delta^{ij}] + \mathrm{vec}[(F^\top (\Delta^{ji})^\top)^\top]
$$

$$
= \mathrm{vec}[F^\top \Delta^{ij}] + \mathrm{vec}[(F^\top \Delta^{ij})^\top]
$$

$$
= \mathrm{vec}[F^\top \Delta^{ij}] + K^{(m,m)} \mathrm{vec}[F^\top \Delta^{ij}]
$$

$$
= (I_{mm} + K^{(m,m)}) \mathrm{vec}[F^\top \Delta^{ij}]
$$

$$
= (I_{mm} + K^{(m,m)})[I_m \otimes F^\top] \mathrm{vec}[\Delta^{ij}].
$$

Thus,

$$
\frac{\partial \mathrm{vec}[F^\top F]}{\partial \mathrm{vec}[F]}
$$
$$
= (I_{mm} + K^{(m,m)})[I_m \otimes F^\top][\mathrm{vec}[\Delta^{11}], \ldots, \mathrm{vec}[\Delta^{m1}], \mathrm{vec}[\Delta^{12}], \ldots, \mathrm{vec}[\Delta^{m2}], \cdots, \mathrm{vec}[\Delta^{1m}], \ldots, \mathrm{vec}[\Delta^{mm}]]
$$
$$
= (I_{mm} + K^{(m,m)})[I_m \otimes F^\top]I_{mm}
$$
$$
= (I_{mm} + K^{(m,m)})[I_m \otimes F^\top].
$$

Combining the first term and second term,

$$\frac{\partial \operatorname{vec}[g(F)]}{\partial \operatorname{vec}[F]}$$

$$= \frac{1}{\|F^\top F\|_{\mathrm{F}} + \epsilon_F} \left( I_{mm} - \frac{1}{(\|F^\top F\|_{\mathrm{F}} + \epsilon_F)\|F^\top F\|_{\mathrm{F}}} \operatorname{vec}[F^\top F] \operatorname{vec}[F^\top F]^\top \right) (I_{mm} + K^{(m,m)})[I_m \otimes F^\top].$$

Since the square commutation matrix is symmetric, using the definition of the commutation matrix,

$$\operatorname{vec}[F^\top F] \operatorname{vec}[F^\top F]^\top K^{(m,m)} = \operatorname{vec}[F^\top F] \operatorname{vec}[F^\top F]^\top (K^{(m,m)})^\top$$

$$= \operatorname{vec}[F^\top F] (K^{(m,m)} \operatorname{vec}[F^\top F])^\top$$

$$= \operatorname{vec}[F^\top F] (\operatorname{vec}[(F^\top F)^\top])^\top$$

$$= \operatorname{vec}[F^\top F] \operatorname{vec}[F^\top F]^\top$$

Therefore,

$$\operatorname{vec}[F^\top F] \operatorname{vec}[F^\top F]^\top (I_{mm} + K^{(m,m)}) = 2 \operatorname{vec}[F^\top F] \operatorname{vec}[F^\top F]^\top.$$

Using this,

$$\frac{\partial \operatorname{vec}[g(F)]}{\partial \operatorname{vec}[F]}$$

$$= \frac{1}{\|F^\top F\|_{\mathrm{F}} + \epsilon_F} \left( I_{mm} - \frac{1}{(\|F^\top F\|_{\mathrm{F}} + \epsilon_F)\|F^\top F\|_{\mathrm{F}}} \operatorname{vec}[F^\top F] \operatorname{vec}[F^\top F]^\top \right) (I_{mm} + K^{(m,m)})[I_m \otimes F^\top]$$

$$= \frac{1}{\|F^\top F\|_{\mathrm{F}} + \epsilon_F} \left( I_{mm} + K^{(m,m)} - \frac{2}{(\|F^\top F\|_{\mathrm{F}} + \epsilon_F)\|F^\top F\|_{\mathrm{F}}} \operatorname{vec}[F^\top F] \operatorname{vec}[F^\top F]^\top \right) [I_m \otimes F^\top]$$

$$= \frac{1}{\|F^\top F\|_{\mathrm{F}} + \epsilon_F} (I_{mm} + K^{(m,m)})[I_m \otimes F^\top] - \frac{2}{(\|F^\top F\|_{\mathrm{F}} + \epsilon_F)^2 \|F^\top F\|_{\mathrm{F}}} \operatorname{vec}[F^\top F] \operatorname{vec}[F^\top F]^\top [I_m \otimes F^\top]$$

$$= \frac{1}{\|F^\top F\|_{\mathrm{F}} + \epsilon_F} (I_{mm} + K^{(m,m)})[I_m \otimes F^\top] - \frac{2}{(\|F^\top F\|_{\mathrm{F}} + \epsilon_F)^2 \|F^\top F\|_{\mathrm{F}}} \operatorname{vec}[F^\top F]([I_m \otimes F] \operatorname{vec}[F^\top F])^\top$$

$$= \frac{1}{\|F^\top F\|_{\mathrm{F}} + \epsilon_F} (I_{mm} + K^{(m,m)})[I_m \otimes F^\top] - \frac{2}{(\|F^\top F\|_{\mathrm{F}} + \epsilon_F)^2 \|F^\top F\|_{\mathrm{F}}} \operatorname{vec}[F^\top F] \operatorname{vec}[FF^\top F]^\top$$

By using this form of $\frac{\partial \operatorname{vec}[g(F)]}{\partial \operatorname{vec}[F]}$ and by defining $R = Q_F \left( G \circ \left( Q_F^\top B^\top \frac{\partial \ell_Y(\mathbf{f}_{X,F,B})}{\partial \mathbf{f}_{X,F,B}} Q_S \right) \right) Q_S^\top S Z_{X,F}^\top$,

$$\nabla_{\operatorname{vec}[F]} L(B, F)$$

$$= \gamma \left( \frac{\partial \operatorname{vec}[g(F)]}{\partial \operatorname{vec}[F]} \right)^\top \operatorname{vec}[R],$$

$$= \gamma \left( \frac{1}{\|F^\top F\|_{\mathrm{F}} + \epsilon_F} (I_{mm} + K^{(m,m)})[I_m \otimes F^\top] - \frac{2}{(\|F^\top F\|_{\mathrm{F}} + \epsilon_F)^2 \|F^\top F\|_{\mathrm{F}}} \operatorname{vec}[F^\top F] \operatorname{vec}[FF^\top F]^\top \right)^\top \operatorname{vec}[R]$$

$$= \gamma \left( \frac{1}{\|F^\top F\|_{\mathrm{F}} + \epsilon_F} [I_m \otimes F](I_{mm} + K^{(m,m)}) - \frac{2}{(\|F^\top F\|_{\mathrm{F}} + \epsilon_F)^2 \|F^\top F\|_{\mathrm{F}}} \operatorname{vec}[FF^\top F] \operatorname{vec}[F^\top F]^\top \right) \operatorname{vec}[R]$$

$$= \left( \frac{\gamma}{\|F^\top F\|_{\mathrm{F}} + \epsilon_F} [I_m \otimes F](I_{mm} + K^{(m,m)}) - \frac{2\gamma}{(\|F^\top F\|_{\mathrm{F}} + \epsilon_F)^2 \|F^\top F\|_{\mathrm{F}}} \operatorname{vec}[FF^\top F] \operatorname{vec}[F^\top F]^\top \right) \operatorname{vec}[R]$$

We compute each of cross terms:

$$[I_m \otimes F] I_{mm} \operatorname{vec}[R] = \operatorname{vec}[FR]$$

$$[I_m \otimes F] K^{(m,m)} \operatorname{vec}[R] = [I_m \otimes F] \operatorname{vec}[R^\top] = \operatorname{vec}[FR^\top]$$

$$\operatorname{vec}[FF^\top F] \operatorname{vec}[F^\top F]^\top \operatorname{vec}[R] = \operatorname{vec}[FF^\top F] \langle F^\top F, R \rangle_{\mathrm{F}}$$

Using these, since vectorization vec is a linear map,

$$\nabla_{\text{vec}[F]} L(B, F)$$

$$= \left( \frac{\gamma}{\|F^\top F\|_{\text{F}} + \epsilon_F} [I_m \otimes F](I_{mm} + K^{(m,m)}) - \frac{2\gamma}{(\|F^\top F\|_{\text{F}} + \epsilon_F)^2 \|F^\top F\|_{\text{F}}} \text{vec}[FF^\top F] \text{vec}[F^\top F]^\top \right) \text{vec}[R]$$

$$= \frac{\gamma}{\|F^\top F\|_{\text{F}} + \epsilon_F} (\text{vec}[FR] + \text{vec}[FR^\top]) - \frac{2\gamma \langle F^\top F, R \rangle_{\text{F}}}{(\|F^\top F\|_{\text{F}} + \epsilon_F)^2 \|F^\top F\|_{\text{F}}} \text{vec}[FF^\top F]$$

$$= \frac{\gamma}{\|F^\top F\|_{\text{F}} + \epsilon_F} \text{vec}[FR + FR^\top] - \frac{2\gamma \langle F^\top F, R \rangle_{\text{F}}}{(\|F^\top F\|_{\text{F}} + \epsilon_F)^2 \|F^\top F\|_{\text{F}}} \text{vec}[FF^\top F]$$

$$= \frac{\gamma}{\|F^\top F\|_{\text{F}} + \epsilon_F} \text{vec}[F(R + R^\top)] - \frac{2\gamma \langle F^\top F, R \rangle_{\text{F}}}{(\|F^\top F\|_{\text{F}} + \epsilon_F)^2 \|F^\top F\|_{\text{F}}} \text{vec}[FF^\top F]$$

Therefore,

$$\nabla_{\text{vec}[F]} L(B, F) = \frac{\gamma}{\|F^\top F\|_{\text{F}} + \epsilon_F} \text{vec}[F(R + R^\top)] - \frac{2\gamma \langle F^\top F, R \rangle_{\text{F}}}{(\|F^\top F\|_{\text{F}} + \epsilon_F)^2 \|F^\top F\|_{\text{F}}} \text{vec}[FF^\top F]$$

$$\nabla_F L(B, F) = \frac{\gamma}{\|F^\top F\|_{\text{F}} + \epsilon_F} F(R + R^\top) - \frac{2\gamma \langle F^\top F, R \rangle_{\text{F}}}{(\|F^\top F\|_{\text{F}} + \epsilon_F)^2 \|F^\top F\|_{\text{F}}} FF^\top F$$

$$= \frac{\gamma}{\|F^\top F\|_{\text{F}} + \epsilon_F} F \left( (R + R^\top) - \frac{2 \langle F^\top F, R \rangle_{\text{F}}}{\|F^\top F\|_{\text{F}}^2 + \epsilon_F \|F^\top F\|_{\text{F}}} F^\top F \right)$$

## C   More on experiments

### C.1   Datasets

**Synthetic chains datasets**   To evaluate the ability of models to capture information from distant nodes, we construct synthetic chains datasets as in Gu et al. [10]. In our experiments, we consider both binary classification and multiclass classification. Assuming the number of classes is $c$, we then have $c$ types of chains and the information of the label class is only encoded as a one-hot vector in the first $c$ dimensions of the node feature vector of the starting end node of the chain. With $c$ classes, $n_c$ chains for each class, and $l$ nodes in each chains, the chain dataset has $c \times n_c \times l$ nodes in total. We choose 20 chains for each class and $c = 5$ for multiclass classification setting. The train set consists 5% nodes while the validation and test set contain 10% and 85% nodes respectively.

**Real-world datasets**   In our real-world experiments, following Pei et al. [21], we use the following real-world datasets:

- **Cornell, Texas and Wisconsin** are web-page graphs of the corresponding universities, where nodes are web pages and edges represent hyper-links between web pages. There are 5 label classes: faculty, student, course, project and staff. These datasets are originally collected by the CMU WebKB project [3]. In our experiments, we use the preprocessed version in Pei et al. [21].
- **Chameleon and Squirrel** are graphs of web pages in Wikipedia of the corresponding topic, originally collected by [22]. We use the labels generated by Pei et al. [21], where the nodes is classified into 5 categories using the amount of their average monthly traffic.

In addition to the above single-graph datasets, we also use Protein-Protein Interaction (PPI) dataset, which contains multiple graphs, to show that EIGNN is applicable to multi-label multi-graph inductive learning setting. PPI dataset has 24 graphs in total and each graph corresponds to a different human issue. In a graph, nodes represents proteins and edges indicates interaction between proteins. Each node can have at most 121 labels, which is originally collected from the Molecular Signatures Database [25] by Hamilton et al. [11]. We also follow the data splits used in [11], i.e., 20 graphs for training, two graphs for validation, and the rest two graphs for testing.

---

[3] http://www.cs.cmu.edu/ webkb/

## C.2 Experimental setting

**Node classification** For node classification task on synthetic and real-world datasets, we mainly choose 9 representative baselines to compare with EIGNN: Graph Convolution Network (GCN) [14], Simple Graph Convolution (SGC) [29], Graph Attention Network [26], Jumping Knowledge Network (JKNet) [32], APPNP [15], GCNII [4], and H2GCN [35]. The presented results are averaged by 20 different runs.

**Noise sensitivity** In synthetic experiments, we add uniform noise $\epsilon \sim \mathcal{U}(-\alpha, \alpha)$ to node features for constructing synthetic datasets with noises. Then, EIGNN and IGNN are trained and evaluated on these datasets with $\alpha = 0.01, 0.1$. The results are averaged by 10 different runs.

In real-world experiments, we compare the robustness of H2GCN, EIGNN, and IGNN against adversarial perturbations on node features. We evaluate the models on evasive setting, i.e., the perturbations are added after the model is trained. We use the trained models of H2GCN, EIGNN, and IGNN with their best performance on Cornell, Texas, and Wisconsin. For each node which is correctly classified, we use two classic methods, i.e., Fast Gradient Sign Method (FGSM) [8] and Projected Gradient Descent (PGD) [18], to add perturbations to node features. For FGSM, we add perturbation $\epsilon \in (0.0001, 0.001, 0.01)$. For PGD, we run 15 iterations with different step sizes. Different combinations of perturbation $\epsilon$ and step size $\alpha$ are used: $(0.01, 0.001), (0.001, 0.0001), (0.0001, 10^{-5})$.

**Hyperparameter setting** To avoid bias, we tune the hyperparameters for each baseline on each real-world dataset. We report the best performance, using the set of hyperparameters which performs the best on the validation set, for each method. For baselines, besides the hyperparameters suggested in their papers, we also conduct a hyperparameter search on learning rate {0.001, 0.05, 0.01, 0.1} and weight decay {5e-4, 5e-6}. For GAT, 8 attention heads are used. For APPNP, 10 propagation layers are used as suggested in [15]. On university datasets (i.e., Cornell, Texas, and Wisconsin), for our model EIGNN, we set the learning rate as 0.5, the weight decay as 5e-6, and $\gamma = 0.8$. On Wikipedia datasets (i.e., Chameleon and Squirrel), we employ batch normalization between the infinite-depth layer and the final linear transformation. After that, the dropout is used with parameter 0.5. The hyperparameter search space is set as follows: learning rate {1e-4, 1e-3, 1e-2}, weight decay {5e-4, 5e-6}.

**Hardware specifications** We run experiments on a machine with Intel(R) Xeon(R) Gold 6240 CPU @ 2.60GHz and a single GeForce RTX 2080 Ti GPU with 11 GB GPU memory.

## C.3 Finite-depth IGNN v.s. Infinite-depth IGNN

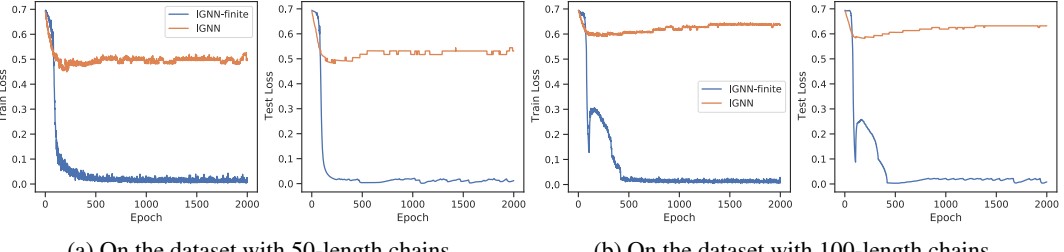

(a) On the dataset with 50-length chains.    (b) On the dataset with 100-length chains.

Figure 4: Train and Test losses versus the number of epochs for IGNN and IGNN-finite on datasets with different chain length.

In Figure 4, we show the train and test loss trajectories of IGNN and IGNN-finite on datasets with different chain lengths. It is clear that IGNN-finite can achieve both lower train and test loss compared with IGNN. For IGNN, the train and test losses both cannot effectively decrease and their trajectories are quite similar, which indicates the issue is that IGNN cannot be effectively learned instead of overfitting problem. The reason why IGNN cannot be effectively learned is that the iterative solver usually generated approximated solutions. To be specific, the error yielded by the iterative solver in the forward pass would be further amplified in the backward pass. Thus, approximation errors make IGNN cannot be optimized well, which leads to the inferior performance.

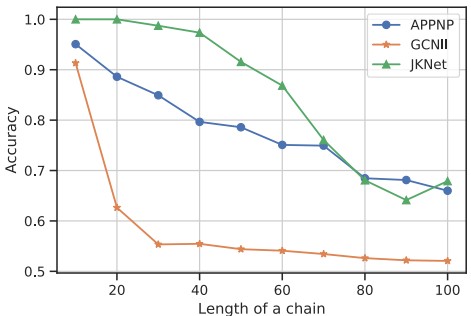

Figure 5: oversmoothness

## C.4 Statistical significance

In our experiments, we conduct multiple runs to get the averaged results. However, for clarity purpose, we omit the error bars on figures represented in Section 5. To test whether the differences are statistically significant, we calculate T-tests for two result series. For Figure 1b, comparing IGNN and EIGNN, the p-values are all smaller than 0.001 for all chain lengths. Comparing IGNN and APPNP (the third-best one), the p-values are all smaller than 0.05 when the chain length is larger than 10. For Figure 1a, comparing IGNN and EIGNN, the p-values are all smaller than 0.05 when the chain length is larger than 20. Comparing IGNN and APPNP (the third-best one), the p-values are all smaller than 1e-4 for all chain lengths. For Figure 2, comparing IGNN and IGNN-fininte, the p-values are smaller than 0.05 when the chain length is larger than 40. Based on the above results, we can conclude the differences are statistically significant.

## C.5 Oversmoothing

A straightforward way to help GNNs capture long-range dependencies is stacking layers. With $T$ hidden layers, GNN models are supposed to capture the information within $T$-hop neighborhoods. However, stacking many layers for traditional GNN models like GCN, SGC, GAT cause oversmoothing. Several works, i.e., APPNP [15], JKNet [32], and GCNII [4], are proposed to mitigate oversmoothing problem. Nevertheless, we demonstrate that they still cannot perfectly capture long-range dependencies. Figure 5 shows the results of APPNP, JKNet and GCNII on datasets with different chain lengths. On the dataset with $T$-length chains, we set the number of propagation layers of these models as $T$. Ideally, with $T$ layers, they should capture the label information from distant nodes and achieve 100% test accuracy. As shown in Figure 5, their performance drops when the chain length increases. The reason might be these model still cannot completely avoid oversmoothing, which makes them ineffectively capture long-range dependencies.

## D  More discussions about limitations

Apart from training, EIGNN conduct eigendecomposition of $S$ which is a one-time preprocessing operation for each graph. The time complexity and memory complexity of a plain full eigendecomposition algorithm are $O(n^3)$ and $O(n^2)$, respectively. It may limit the applicability of EIGNN on some cases where the complexity of this preprocessing is of importance. However, we can consider using truncated eigendecomposition (i.e., use top k eigenvalues and the corresponding eigenvectors) to reduce the time and memory complexity. As the natural sparsity of graphs, using truncated eigendecomposition is reasonable.

Furthermore, another possible solution to mitigate the cost of eigendecomposition of $S$ is to use graph coarsening to reduce the size of graph or graph partition to partition a large graph into several small graphs without losing much information. Several works have proposed on this topics [3, 12]. Worth note that graph partition is also used in a well-known work Cluster-GCN [5] which focuses on scaling up GCNs to a large graph with millions nodes. Cluster-GCN splits a given graph into k non-overlapping subgraphs. Therefore, we believe that graph partition is indeed a potential solution for mitigating the scalability problem in our work. How to scale up implicit graph model for large

graphs is an interesting topic to explore in the future. We leave detailed analyses and empirical experiments for future research.

Another potential limitation is that EIGNN can be slow when the number of feature dimensions is very large (e.g., 1 million), which is rare in practice. Future work could propose a new model which allows for efficient training even with large feature dimension.