# OpenReview forum: "EIGNN: Efficient Infinite-Depth Graph Neural Networks"
_NeurIPS.cc/2021/Conference — NeurIPS 2021 Poster_

### Official Review · Reviewer_emSJ · 2021-07-14

**Rating:** 6
**Confidence:** 4

**Summary:**

The authors propose a linear infinite-depth GNN, whose forward pass is a convergent series whose limit can be written in closed-form. Since the series involves the powers of the normalised adjacency matrix, the resulting model captures interactions between all pairs of connected nodes. At the same time, the existence of a closed-form expression makes the gradients easy to compute and provides an easy way to train the model. The authors show how the computational complexity of the model can be improved by diagonalising some of the matrices involved in the computation. The paper validates the hypothesis that the model can capture long-range interactions better than existent models in a synthetic setting and also on small real-world heterophilic graphs.

**Limitations And Societal Impact:**

The authors have not addressed this in sufficient detail.

**Main Review:**

## Strong points

- The authors introduce a model that has many attractive properties: (1) It has residual connections, which have been shown to play an important role in a graph setting, (2) it involves powers of the adjacency matrix which in the limit allows all connected pairs of nodes to interact (3) The series has a nice closed-form expression in the limit that allows the model to be easily trained, (3) the model does not require solvers, which are generally slow or suffer from instabilities.
-  The only inconvenience of the model is that the complexity of computing the closed-form expression is not practical. The authors also offer a (partial) solution for this by proposing various approximations based on the eigendecomposition of certain matrices.
- The chain dataset is a great synthetic setting for testing the theoretically justified ability of the model to capture long-range interactions and the authors compare the model carefully against IGNNs in this setting. They also perform a reasonably convincing ablation for IGNN to better understand its limitations.
- The authors also consider a real-world setting formed of heterophilic graphs, which is indeed an appropriate class of graphs for testing to some extent the long-range interactions that the model can capture.

## Improvements and Questions

- Some of the motivations behind the work are slightly incorrectly stated and do injustice to classic GNNs. Therefore, these could be rephrased:
  - In the abstract (and other places) the authors claim that "finite layers" -> "unable to capture long-range interactions". This is not correct. For all practical purposes, the long-range interactions that can be found in practice are bounded by some constant $L$ and a GNN with $L$ layers will be able to capture them.
  - In line 34 "GNN models with numerous layers require excessive computational cost in practice". The complexity of a message passing GNN with $k$ layers in a graph with $E$ edges is $\mathcal{O}(kE)$. Even assuming that $k ~ \mathcal{O}(V)$ (the number of vertices), this is still $\mathcal{O}(VE)$, which I would not call "excessive". Note that for the eigendecomposition of $S$ in line 176, the complexity is $\mathcal{O}(V^3)$, which is worse when the graph is sparse (I also have a question about this below).
  - In line 30: "... poor empirical performance when stacking more than a few layers, which has been referred to as oversmoothing". Oversmoothing is only one of the causes of why Deep GNNs often have problems and it is more common among convolutional models (https://arxiv.org/abs/1905.10947). There is also over squashing (https://arxiv.org/abs/2006.05205) and it is also likely related to optimisation issues as well.
- Regarding the complexity of the method, the $\mathcal{O}(V^3)$ eigendecomposition of $S$ from line 176 does not seem to be included in the reported computational complexity. Why is that the case? That is very problematic for many applications where the number of nodes is well beyond $1$ million. Is this why the authors evaluate only on extremely small graphs (see my next point)? The authors might argue that the eigendecomposition of $S$ is not such a big problem because it is a one-time pre-processing step, but this could still limit its applicability in a real-world setting where the graph is often changing in time.
- As mentioned above, the real-world benchmarks used by the authors are extremely small. While including small-sized graphs is totally justified, I would expect to see at least some medium-sized graphs in the evaluation as well (e.g. from https://ogb.stanford.edu/ or https://arxiv.org/abs/2003.00982).  Based on this and my point above, I have doubts about the scalability of the model in a practical setting.
- Another limitation that is not mentioned is that using the eigendecomposition of $S$ in the model makes the model inapplicable to multiple graphs because the eigenbasis changes from one graph to another and even for isomorphic graphs, it is not uniquely defined (for more details see e.g. https://arxiv.org/abs/1611.08097). This can be a problem for many practical settings where we want to classify nodes from multiple graphs or maybe we want to perform pre-training on another graph.
- The limitations of the model have not been properly addressed. The authors have included only a sentence in the conclusion on this. For instance, my point on the $\mathcal{O}(V^3)$ complexity or the transferability across different graphs could have been mentioned. Are there any other limitations that the authors could mention?
- All the figures in the experimental section are missing standard deviations. All experiments should be run over multiple weight initialisations for all the models. Without error bars around the curves in the figures, the reported results could just be lucky runs.
- While I appreciate the experiments on noise robustness, they only show results on a synthetic graph where the proposed model was already performing better in the first place. At the same time, no justification is offered as to why the proposed model would be more robust to noise. An intuitive justification (if not theoretical) should be presented and validated at least.
- The paper does not comment at all on the closed-form expression from Proposition 1 to provide a deeper understanding. The expression looks very interesting and the authors could attempt to provide an intuition about what it actually does. For instance, how is the information from different hops around a source node weighted by this expression?
- A Question: What is the purpose of Proposition 3 and Equation (9)? Since the forward pass has a trivial closed-form expression, the gradient can be trivially computed by an automatic differentiation package. Furthermore, the authors do not seem to use the gradient in their follow-up analysis and, therefore, it could be simply added to the appendix since it does not seem to provide any essential information to the reader. Or is the computation of the derived gradient more efficient than what automatic differentiation would compute? I do not believe that is the case, but if it is, then it should be proven.

EDIT: After the rebuttal, I have increased my score from 5 to 6.

**Time Spent Reviewing:**

3

---

> ### Author Response · Authors · 2021-08-10
> **Response to all concerns raised by the reviewer (Part 1)**
>
> [Response split in two posts: 1/2] We thank Reviewer emSJ for the valuable feedback and recommendations for improving the manuscript. We addressed all the concerns as described below.
>
> For the sentence suggested to be rephrased, we will rephrase them as suggested.
>
> 1. "Regarding the complexity of the method, the O(V3) eigendecomposition of S from line 176 does not seem to be included in the reported computational complexity. Why is that the case? That is very problematic for many applications where the number of nodes is well beyond 1 million. Is this why the authors evaluate only on extremely small graphs (see my next point)? The authors might argue that the eigendecomposition of S is not such a big problem because it is a one-time pre-processing step, but this could still limit its applicability in a real-world setting where the graph is often changing in time."
>
> We agree that eigendecompositon for a large graph with more than 1 million nodes would be slow. However, in this case, we can consider using the truncated eigendecomposition (only consider top k eigenvalue) for adjacency matrix S to reduce the computational and memory complexity in EIGNN. Furthermore, another possible solution to mitigate it is to use graph coarsening to reduce the size of graph or graph partition to partition a large graph into several small graphs without losing much information. We have seen several works on this topics [1,2,3]. Worth note that graph partition is also used in a well-known work Cluster-GCN [5] which focuses on scaling up GCNs to a large graph with millions nodes. Cluster-GCN splits a given graph into k non-overlapping subgraphs. It uses METIS graph partition algorithm [6]. Therefore, we believe that graph partition is indeed a potential solution for mitigating the scalability problem in our work. How to scale up implicit graph model for large graphs is an interesting topic to explore in the future. For a real-world setting where the graph is changing or multiple graphs exist, we discuss in details in a later response.
>
> 2. "As mentioned above, the real-world benchmarks used by the authors are extremely small. While including small-sized graphs is totally justified, I would expect to see at least some medium-sized graphs in the evaluation as well (e.g. from https://ogb.stanford.edu/ or https://arxiv.org/abs/2003.00982). Based on this and my point above, I have doubts about the scalability of the model in a practical setting."
>
> For the experiments on medium-sized graphs, we conduct a simple experiment on ogbn-arxiv dataset with 169k nodes. We only use top 10k eigenvalues and their corresponding eigenvectors. EIGNN can achieve 71.40% accuracy on test set. We note that the performance is not that appealing. But considering we only use 1/10 eigen vectors, we verify that truncated eigendecomposition is also a potential solution for mitigating the scalability problem. As the natural sparsity of graphs, using truncated eigendecomposition is reasonable.
>
> 3. "Another limitation that is not mentioned is that using the eigendecomposition of S in the model makes the model inapplicable to multiple graphs because the eigenbasis changes from one graph to another and even for isomorphic graphs, it is not uniquely defined (for more details see e.g. https://arxiv.org/abs/1611.08097). This can be a problem for many practical settings where we want to classify nodes from multiple graphs or maybe we want to perform pre-training on another graph."
>
> For multiple-graphs setting, if the multiple graphs are fixed, we can also save the results of eigendecomposition of S from different graphs when we first process the graph and re-use them later for training and also inference. If multiple graphs keep evolving, the problem becomes graph neural networks on temporal graphs, which is not the focus of this work. However, we agree that how we can have infinite-depth GNNs working on temporal graphs is a very interesting point for future works. Pre-training on another graph is similar with the setting on fixed multiple graphs. We can also save and reuse the results of eigendecomposition of S from both the pretrain graph and fine-tuning graphs. It is also similar with directly change the adjacency matrix for GCN under inductive setting (see https://github.com/tkipf/gcn/issues/79). In our case, we change the eigenbasis when graphs are different. But all these results only need to be computed once.  To verify this, we conduct another set of experiments on PPI dataset as in IGNN paper. PPI dataset contains multiple graphs and ~57k nodes in total. We show the results as follows:
>
> |   | Test Micro f1   |      Training time per epoch      |  Precompute time for Eigendecomposition of all S (only once) |
> |----------|----------|-------------|------|
> | EIGNN |  97.97 | 2.23s | 40.59s |
> | IGNN |    97.6   |   35.03s | N.A.|
>
> We use the test micro f1 of IGNN reported in their paper and rerun their code to get the running time. The number of epochs during training is more than 1000 as suggested in IGNN. Therefore, on PPI dataset, the total running time of EIGNN is much less than IGNN and the precompute time for eigendecomposition of S can be neglected. Meanwhile, EIGNN can achieve even slightly better performance. These experiments verify that EIGNN is applicable to multi-graph inductive setting.
>
> 4. "The limitations of the model have not been properly addressed. The authors have included only a sentence in the conclusion on this. For instance, my point on the O(V3) complexity or the transferability across different graphs could have been mentioned. Are there any other limitations that the authors could mention?"
>
> We agree that the one-time preprocessing step with $O(V^3)$ complexity is a limitation of EIGNN. But we believe the transferability across different graphs is not a limitation as we explain above. We will revise our future version based on your valuable feedbacks.
>
> 5. "All the figures in the experimental section are missing standard deviations. All experiments should be run over multiple weight initialisations for all the models. Without error bars around the curves in the figures, the reported results could just be lucky runs."
>
> Sorry for not providing standard deviations on figures. However, we did run the experiments over multiple weight initializations. Figure 1 and Figure 2 are generated by averaging the results obtained from 20 different runs with different random seeds. Figure 3 is generated by averaging the results from 10 different runs, which is stated in the appendix.
>
> 6. "While I appreciate the experiments on noise robustness, they only show results on a synthetic graph where the proposed model was already performing better in the first place. At the same time, no justification is offered as to why the proposed model would be more robust to noise. An intuitive justification (if not theoretical) should be presented and validated at least."
>
> We agree that EIGNN already perform best in the first place on synthetic graphs, and it would be not directly strongly support that EIGNN is more robust to noise on synthetic graphs.
>
> However, in addition to the experiments on noise robustness on synthetic graphs, we also provide results about adversarial perturbation for features on real-world graphs. Note that, in the first place (no perturbation setting), the gap between H2GCN and EIGNN is much smaller than the gap in perturbated setting when H2GCN and EIGNN under a same amount of perturbations. For example, using FGSM with 0.001 perturbation, H2GCN only achieves ~27% accuracy while EIGNN still has ~84% accuracy.
>
> For the justification, intuitively, we think EIGNN is not/less sensitive to local neighborhoods while finite message passing GNNs could be very/more sensitive to local neighborhoods. Finite message passing GNNs usually only have a few layers and the node representations could be changed a lot when aggregating from perturbated node features. And this can make those models not robust to noise/perturbations. In contrast, EIGNN is an infinite-depth GNN model, which can consider more about global information and become less sensitive to perturbations from local neighborhoods.
>
> 7. "The paper does not comment at all on the closed-form expression from Proposition 1 to provide a deeper understanding. The expression looks very interesting and the authors could attempt to provide an intuition about what it actually does. For instance, how is the information from different hops around a source node weighted by this expression?"
>
> We thank the reviewer for the valuable feedback. The closed-form expression is derived from Equation (2)-(4). The weights in the closed-form expression are learned from the data. For the expression itself, we have not had an intuition about that. We appreciate the valuable insight and we would consider more about this as a future direction.

---

> > ### Author Response · Authors · 2021-08-10
> > **Response to all concerns raised by the reviewer (Part 2)**
> >
> > [Response split in two posts: 2/2]
> >
> > 8. "A Question: What is the purpose of Proposition 3 and Equation (9)? Since the forward pass has a trivial closed-form expression, the gradient can be trivially computed by an automatic differentiation package. Furthermore, the authors do not seem to use the gradient in their follow-up analysis and, therefore, it could be simply added to the appendix since it does not seem to provide any essential information to the reader. Or is the computation of the derived gradient more efficient than what automatic differentiation would compute? I do not believe that is the case, but if it is, then it should be proven."
> >
> > Thanks for the suggestion. We agree that Equation (9) is a bit redundant. We will revise it and add it to the appendix in the future version. However, we believe that Proposition 3 and Equation (8) should be in the main text. Because presenting them can help us introduce the improved version in Section 4.2 (especially Equation (11) and (12)). And it can provide a clear comparison between the trivial version and the improved version. Equation (11) is an improvement using eigendecomposition compared with Equation (8), and Equation (12) is an improvement via avoiding $\frac{\partial \operatorname{vec}[g(F)]}{\partial \operatorname{vec}[F]}$ compared with Eq (11). Worth note that an autograd framework can only autocompute Equation (11). In Equation (12), we show that we can reduce the memory cost further, which cannot be done by an autograd package, and we provide the rigorous proof for it in the appendix. In Section 4.2, we omit the gradient of B since it indeed can be trivially computed by an autograd framework, which we state in Line 190.
> >
> > References
> >
> > [1] Graph Coarsening with Neural Networks, ICLR 2021
> >
> > [2] GRAPH PARTITION NEURAL NETWORKS FOR SEMI-SUPERVISED CLASSIFICATION, https://arxiv.org/abs/1803.06272
> >
> > [3] Scaling Up Graph Neural Networks Via Graph Coarsening, KDD 2021
> >
> > [4] GEOM-GCN: GEOMETRIC GRAPH CONVOLUTIONAL NETWORKS, ICLR 2020
> >
> > [5] Simple and Deep Graph Convolutional Networks, ICML 2020
> >
> > [6] Cluster-GCN: An Efficient Algorithm for Training Deep and Large Graph Convolutional Networks, KDD 2019
> >
> > [7] METIS: Serial Graph Partitioning and Fill-reducing Matrix Ordering, http://glaros.dtc.umn.edu/gkhome/views/metis

---

> > > ### Comment · Reviewer_emSJ · 2021-08-16
> > > **Response to Authors**
> > >
> > > Thank you for your detailed reply to my questions and comments! Please find my response below.
> > >
> > > 1. I still think the paper has not sufficiently investigated the engineering aspects regarding the $O(V^3)$ eigendecomposition. The analysis using a truncated number of eigenvectors is indeed a step in the right direction, but much more work is needed to understand the tradeoffs of this approximation and others. For instance, the paper could have explored what is the best strategy for choosing the eigenvectors. Should it be the lowest-frequencies or a mix of low and high-frequency, etc? Additionally, I think the alternative methods like the graph coarsening should have also been explored in detail. Without these, I am not convinced by the practical utility of the model. Long-range interactions are particularly important in large graphs and that is an area where the computational aspects have been neglected.
> > > 2. I appreciate the experiment on ogbn-arxiv that the authors have run. While the result is not close to SOTA, I would be more interested to see how it compares with the main baselines you consider in the paper (e.g. IGNN)? Additionally, I would have liked to see many more large benchmarks. For instance, the SBM node-classification benchmarks (PATTERN, CLUSTER) from the Benchmarking GNNs paper could have been run at least. OGB also contains more additional node-level benchmarks. Studying various approximations from point 1) in this setting seems essential to me.
> > > 3. I am not sure if my point regarding the fact that the eigenbasis is not uniquely defined has been understood. Consider two isomorphic graphs. First of all, their eigenvectors are defined only up to a change of sign ($\pm 1$). Additionally, suppose there is an eigenvalue with multiplicity higher than one. Then the eigenspace corresponding to that eigenvalue is defined up to a rotation. The fact one might obtain different eigenvectors even for the same graph could cause instabilities in the model and limit its applicability to multiple graphs. Please correct me if there is something wrong with my reasoning here! I also appreciate the additional experiment on the PPI dataset!
> > > 5. Will you add the error bars to the figures? Are the differences statistically significant?
> > >
> > > I am satisfied with the response from the other points, which I did not mention above.

---

> > > > ### Author Response · Authors · 2021-08-23
> > > > **Response to the reviewer**
> > > >
> > > > Thank you for the further feedback and suggestions. We post our response below.
> > > >
> > > >
> > > > 1. Regarding the strategy for choosing eigenvectors in truncated eigendecomposition, we think using largest values is more reasonable than a mix of largest and smallest ones. The reason is that the smallest eigenvalues of a normalized adjacency matrix are always close to zero, which are not that informative. 0 can be proved to be the smallest eigenvalue of a normalized adjacency matrix (see the supplementary material of https://arxiv.org/pdf/1902.07153.pdf). Empirically, the strategy of using a mix of largest and smallest eigenvalues consistently performs badly (less than 70% accuracy), which is worse than directly using largest eigenvalues. Based on the above reasoning and empirical results, we think it is more natural and reasonable to use the largest eigenvalues when we consider truncated eigendecomposition. We appreciate the suggestion again; we will revise and include this part in the future version of our manuscript.
> > > >
> > > > 2. As suggested, we conduct experiments for IGNN on ogbn-arxiv dataset. It achieves 71.08% accuracy, which is even worse than the result of our method using truncated eigendecomposition as we reported before. We also conduct additional experiments for EIGNN and IGNN on PATTERN dataset. The results show that IGNN performs slightly better than EIGNN (88.71 v.s. 88.89). Note that in PATTERN, we do not need truncated eigendecomposition as it is a multiple-graph dataset where each graph is not large. Due to limited time, we leave more experiments on mentioned benchmarks as future work.
> > > >
> > > > 3. We agree that eigenvectors are defined up to a change of sign. But we think it would not cause instabilities and limit the applicability to multiple graphs. Note that in our model, we use eigenvectors and eigenvalues to recover the adjacency matrix. As long as the adjacency matrix is informative, our model should work similarly. For two isomorphic graphs, even the adjacency matrices are different, the generated representations should be similar. Thus, we think our model would not face instability.
> > > >
> > > > 4. We will add error bars to the figures in the future version of our paper. To test whether the differences are statistically significant, we calculate T-tests for two result series. For Figure 3, comparing IGNN and EIGNN, the p-values are all smaller than 0.001 for all chain lengths. Comparing IGNN and APPNP (the third-best one), the p-values are all smaller than 0.05 when the chain length is larger than 10. For Figure 2, comparing IGNN and EIGNN, the p-values are all smaller than 0.05 when the chain length is larger than 20. Comparing IGNN and APPNP (the third-best one), the p-values are all smaller than 1e-4 for all chain lengths. For Figure 4, comparing IGNN and IGNN-fininte, the p-values are smaller than 0.05 when the chain length is larger than 40. Based on the above results, we believe the differences are statistically significant.
> > > >
> > > > 5. We would like to emphasize that our work focuses more on how to avoid approximation errors from iterative solvers used in previous implicit graph models. We think it can be a valuable step on the topic of implicit graph models. For the engineering aspects regarding the complexity of eigendecomposition, besides our response above, we would like to point out that a normalized adjacency matrix must be a positive semi-definite matrix. This property makes that the eigendecomposition can be recast by an SVD (see https://en.wikipedia.org/wiki/Singular_value_decomposition#Relation_to_eigenvalue_decomposition). Note that there are some much more efficient approximated SVD algorithms whose complexity is $O(\text{nnz}(S)k)$ for k-truncated decomposition [1,2]. It can be used in our work as a future direction as well.
> > > >
> > > > Thank you for the valuable feedback again. Hope our response can convince you and hope you can raise your score if you are satisfied with our response.
> > > >
> > > > References:
> > > >
> > > > [1] Randomized Block Krylov Methods for Stronger and Faster Approximate Singular Value Decomposition, NIPS 2015
> > > >
> > > > [2] Low Rank Approximation and Regression in Input Sparsity Time, STOC 2013.

---

> > > > > ### Comment · Reviewer_emSJ · 2021-08-30
> > > > > **New response**
> > > > >
> > > > > Thank you for your response! I think most of my comments have been sufficiently addressed and I will therefore raise my score to 6.

---

> > > > > > ### Author Response · Authors · 2021-08-30
> > > > > > **Thank you!**
> > > > > >
> > > > > > Thank you for raising your score! And we really appreciate all the helpful suggestions you provided during this phase. We will improve the future version of our paper according to your suggestions.

---

### Official Review · Reviewer_9hrH · 2021-07-16

**Rating:** 6
**Confidence:** 4

**Summary:**

This paper proposes an efficient method for infinite-depth GNN model that allows learnable transformations in each layer. A more efficient computation (eigen-decomposition) of the close-form solution is discussed and proved in the paper. In the experiments, EIGNN shows superior performance on both synthetic and real datasets which requires long-range dependencies capture. The robustness of the method is evaluated by perturbation of node features as well.

**Limitations And Societal Impact:**

Yes.

**Main Review:**

Overall, the paper is clear and easy to follow. The result presented in the paper is both significant and efficient, which aligns well with the original claims made in the paper.

Weakness and Questions:
1. Some of the interesting work at long-range dependencies are neglected in both related work and experiments. For example, the author mentions APPNP/PPNP in the experiment, while I think this one is quite related with the main contribution of the paper and author should discuss this one in related work. Also, another work called "Continuous Graph Neural Networks" [1] is worth discussing in the paper. In its 4.2, it also applies learnable parameters in an infinite-layer GNNs and conduct solution. The author should consider discuss and compare with this method as well.
2. The dataset selected might be biased towards to the long-range dependencies on node classification. I notice the proposed method yield a large margin to the selected dataset. However, these datasets are both small and not well-known for node classification (see OGB for large benchmarks). Why not you do graph classification instead like the original IGNN paper?
3. The scalability of the method. As far as I understand, although eigen-decomposition is way more efficient than inverse computation. But in a large network with millions nodes, can you still handle it via eigen-decomposition?

[1] Xhonneux, Louis-Pascal, Meng Qu, and Jian Tang. "Continuous graph neural networks." International Conference on Machine Learning. PMLR, 2020.

**Time Spent Reviewing:**

3

---

> ### Author Response · Authors · 2021-08-10
> **Responses to all concerns raised by the reviewer**
>
> We thank Reviewer 9hrH for the valuable feedback and recommendations for improving the manuscript. We addressed all the concerns as described below.
>
> 1.	"Some of the interesting work at long-range dependencies are neglected in both related work and experiments. For example, the author mentions APPNP/PPNP in the experiment, while I think this one is quite related with the main contribution of the paper and author should discuss this one in related work. Also, another work called "Continuous Graph Neural Networks" [1] is worth discussing in the paper. In its 4.2, it also applies learnable parameters in an infinite-layer GNNs and conduct solution. The author should consider discuss and compare with this method as well."
>
> We thank the reviewer for the valuable suggestion. We agree that APPNP/PPNP is related to this work. However, we do discuss it in the paper. Without discussing the details, we simply cite APPNP/PPNP as one of previous GNN models in related work. In the first paragraph of Section 4.3, we discuss APPNP/PPNP in details and compare it with EIGNN.
> For the work “Continuous Graph Neural Network”, it requires the eigendecomposition for adjacency matrix as well. In addition, it requires matrix logarithm on adjacency matrix, which is intractable to compute. To get rid of matrix logarithm, it uses the first order Taylor approximation $\ln A \approx (A – I)$, which is inaccurate as it can introduce much approximation error. We will consider discussing and comparing with it in the future version.
>
> 2.	"The dataset selected might be biased towards to the long-range dependencies on node classification. I notice the proposed method yield a large margin to the selected dataset. However, these datasets are both small and not well-known for node classification (see OGB for large benchmarks). Why not you do graph classification instead like the original IGNN paper?"
>
> As the main claim of our work is that EIGNN can capture the long-range dependencies on graph while many existing works are not able to do so, we select the datasets which are heterophilic and can support our claim. These datasets are also commonly used for node classification in existing works related to long-range dependencies (e.g., [4,5]). For graph classification, in our humble opinion, we think there is no clear/direct connection between graph classification task and long-range dependencies. However, we conduct another set of experiments on PPI dataset as in IGNN paper. PPI dataset contains multiple graphs and ~57k nodes in total. We conduct the experiments on it and compare the efficiency between EIGNN and IGNN. We show the resuls on PPI dataset as below:
>
> |   | Test Micro f1   |      Training time per epoch      |  Precompute time for Eigendecomposition of all S (only once) |
> |----------|----------|-------------|------|
> | EIGNN |  97.97 | 2.23s | 40.59s |
> | IGNN |    97.6   |   35.03s | N.A.|
>
> We use the test micro f1 of IGNN reported in their paper and rerun their code to get the running time. The number of epochs during training is more than 1000 as suggested in IGNN. Therefore, on PPI dataset, the total running time of EIGNN is much less than IGNN and the precompute time for eigendecomposition of S can be neglected. Meanwhile, EIGNN can achieve even slightly better performance. In addition, the experiments verify that EIGNN is applicable to multi-graph inductive setting.
>
> 3.	"The scalability of the method. As far as I understand, although eigen-decomposition is way more efficient than inverse computation. But in a large network with millions nodes, can you still handle it via eigen-decomposition?"
>
> For a single large graph with millions nodes, the full eigendecomposition would be very slow and the memory cost would be high. However, in this case, we can consider using the truncated eigendecomposition (only consider top k eigenvalue) for adjacency matrix S to reduce the computational and memory complexity in EIGNN. Furthermore, another possible solution is to use graph coarsening to reduce the size of graph or graph partition to partition a large graph into several small graphs without losing much information. We have seen several works on this topics [1,2,3]. Worth note that graph partition is also used in a well-known work Cluster-GCN [6] which focuses on scaling up GCNs to a large graph with millions nodes. Cluster-GCN splits a given graph into k non-overlapping subgraphs. It uses METIS graph partition algorithm [7]. Therefore, we believe that graph partition is indeed a potential solution for mitigating the scalability problem in our work. How to scale up implicit graph model for large graphs is an interesting topic to explore in the future.
>
> In addition, we conduct a simple experiment on ogbn-arxiv dataset with 169k nodes. We only use top 10k eigenvalues and their corresponding eigenvectors. EIGNN can achieve 71.40% accuracy on test set. We note that the performance is not that appealing. But considering we only use 1/10 eigen vectors, we verify that truncated eigendecomposition is also a potential solution for mitigating the scalability problem. As the natural sparsity of graphs, using truncated eigendecomposition is reasonable.
>
>
> References
>
> [1] Graph Coarsening with Neural Networks, ICLR 2021
>
> [2] GRAPH PARTITION NEURAL NETWORKS FOR SEMI-SUPERVISED CLASSIFICATION, https://arxiv.org/abs/1803.06272
>
> [3] Scaling Up Graph Neural Networks Via Graph Coarsening, KDD 2021
>
> [4] GEOM-GCN: GEOMETRIC GRAPH CONVOLUTIONAL NETWORKS, ICLR 2020
>
> [5] Simple and Deep Graph Convolutional Networks, ICML 2020
>
> [6] Cluster-GCN: An Efficient Algorithm for Training Deep and Large Graph Convolutional Networks, KDD 2019
>
> [7] METIS: Serial Graph Partitioning and Fill-reducing Matrix Ordering, http://glaros.dtc.umn.edu/gkhome/views/metis

---

> > ### Comment · Reviewer_9hrH · 2021-08-25
> > **Thanks for your response and I keep my score.**
> >
> > Thanks for your detailed explanations. I strongly suggest you add method like APPNP into table 2 as well (to be most consistent with your synthetic experiment 1). I understand your method should be very good on graph with long-range dependencies as shown in table 2. I appreciate the author for providing result on larger benchmarks. I think it's interesting to see how different method perform under a general graph dataset like ogbn-arxiv or ogbn-product (comparison between EIGNN and other infinite or higher-order GNN as mentioned in the paper). Such a result is more convincing to show the model is compromising the performance on more general datasets.

---

> > > ### Author Response · Authors · 2021-08-30
> > > **New Response**
> > >
> > > We really appreciate your suggestions. However, we would like to point out that we have already provided the results of APPNP in table 2 as shown in our submission. All methods in the synthetic experiments are also presented in table 2.
> > >
> > > As suggested, we conducted experiments for IGNN on ogbn-arxiv dataset. It achieves 71.08% accuracy, which is even worse than the result of EIGNN using truncated eigendecomposition as we reported before. We will include this part in the future version of our paper. Due to limited time, we leave empirical comparisons between EIGNN and other infinite-depth GNNs on more benchmarks as future work.
> > >
> > > Again, thank you for your helpful suggestions during this phase! And hope our response can convince you.

---

### Official Review · Reviewer_mzDR · 2021-07-17

**Rating:** 5
**Confidence:** 4

**Summary:**

The paper proposes an infinite-depth GNN which captures long-range dependencies in the graph while avoiding iterative solvers by deriving a closed-form solution. The eigendecomposition method is introduced to further improve efficiency. Experiments on several datasets shows improved performance.



**Limitations And Societal Impact:**

The limitation of EIGNN should be discussed.

**Main Review:**

1. The paper proposes the EIGNN in Section 4. The model is formulated in Equations (2), (3), and (4) but didn't provide any explanation for this model itself. Therefore, the motivation of the model is unclear.

2. Instead, Section 4 focuses on introducing the forward and backward computation. In section 4.1, it is clear that the computation and memory cost is not acceptable because it requires dealing with the inverse of a very large matrix and the inversed matrix is dense.

3. In section 4.2, an eigendecomposition method is introduced to improve efficiency. However, it still requires the eigendecomposition on the matrix $S$ whose size is the number of nodes. This is still very expensive for a reasonably large graph, and the result of the decomposition are dense matrices that require significantly large memory and costly computation.

4. The paper tries to avoid iterative solvers that cause approximation errors. However, there is no convincing evidence to show how the approximation error impacts the performance. Therefore, the advantages of using a closed-form solution are unclear while the drawbacks are obvious. More ablation study would be helpful.

5. The datasets in the experiments are very small graphs, it is unclear how the model performs in large graphs in terms of both efficiency and effectiveness. The efficiency comparison with iterative solvers will be helpful.

6. In the experiments on the synthetic chain data, there is a huge performance gap between EIGNN and IGNN. Considering that IGNN is able to capture long-range dependencies, it is unclear where the improvements of EIGNN come from. In the experiments on real-world data, the performance gap between EIGNN and H2GCN is very small while the gap between EIGNN and IGNN is huge. It is a bit confusing and again raises the concern -- where do the improvements of EIGNN come from.

7. While it is possible to verify the benefits of capturing long-range dependencies in the synthetic graph, it is unclear whether this property helps in real-world graphs and how to confirm it.

Overall, this paper proposes an infinite-depth GNN with a closed-form solution. However, the computation and memory cost is still very high. Moreover, without an explanation for the model, the reason for the performance improvement is unclear.

--------------------------
During the discussion, I have raised my rating from 3 to 5.

**Time Spent Reviewing:**

3

---

> ### Author Response · Authors · 2021-08-10
> **Responses to all concerns raised by the reviewer**
>
> We thank reviewer mzDR for the valuable feedback and recommendations for improving the manuscript. However, we respectfully disagree with some critical points raised by the reviewer. In our humble opinion, we believe there are some misunderstandings or some important parts of our paper are missed, and we hope to convince her/him about the validity of our claims.
>
> 1.	"The paper proposes the EIGNN in Section 4. The model is formulated in Equations (2), (3), and (4) but didn't provide any explanation for this model itself. Therefore, the motivation of the model is unclear."
>
> We briefly state the motivations (Line 126-128) in the paper. We explain the motivation of Equation (2) – (4) as follows: 1) we want residual connections, which are also used in finite GNN models (i.e., APPNP and GCNII) and have been shown important in graph learning setting.
> 2)	Learnable weights for propagation on graphs (i.e., g(F)). In contrast, APPNP only directly propagates information without learnable weights.
> 3)	Restrict $\gamma$ to make sure the convergence of this infinite sequence.
>
> 2.	"Instead, Section 4 focuses on introducing the forward and backward computation. In section 4.1, it is clear that the computation and memory cost is not acceptable because it requires dealing with the inverse of a very large matrix and the inversed matrix is dense."
>
> In fact, we are not very sure about the meaning of this point. We assume that the point is saying section 4.1 is redundant. The purpose of Section 4.1 is to show that we can first obtain a trivial closed-form solution for the infinite sequence without iterative solvers. In contrast, IGNN heavily relies on iterative solvers. The memory cost of this closed-form solution is huge and that is also the reason why we introduce the eigendecomposition method to further reduce the memory cost in section 4.2. Without section 4.1, it is hard to understand why we can have the improved version by eigendecomposition.
>
> 3.	"In section 4.2, an eigendecomposition method is introduced to improve efficiency. However, it still requires the eigendecomposition on the matrix S whose size is the number of nodes. This is still very expensive for a reasonably large graph, ...".
>
> For a single large graph with millions nodes, the full eigendecomposition would be slow and the memory cost would be high. However, in this case, we can consider using the truncated eigendecomposition (only consider top k eigenvalue) for adjacency matrix to reduce the computational and memory complexity in EIGNN. Furthermore, another possible solution is to use graph coarsening to reduce the size or graph partition to partition a large graph into several small graphs. Several works focus on this topics [1,2,3]. Worth note that graph partition is also used in a well-known work Cluster-GCN [4] which focuses on scaling up GCNs to a large graph with millions nodes. Cluster-GCN splits a given graph into k non-overlapping subgraphs using METIS graph partition algorithm [5]. Therefore, we believe that graph partition is indeed a potential solution for mitigating the scalability problem in our work. How to scale up implicit graph model for large graphs is an interesting topic to explore in the future.
>
> In addition, we conduct a simple experiment on ogbn-arxiv dataset with 169k nodes. We only use top 10k eigenvalues and their corresponding eigenvectors. EIGNN can achieve 71.40% accuracy on test set. We note that the performance is not that appealing. But considering we only use 1/10 eigen vectors, we verify that truncated eigendecomposition is also a potential solution for mitigating the scalability problem. As the natural sparsity of graphs, using truncated eigendecomposition is reasonable.
>
> 4.	"The paper tries to avoid iterative solvers that cause approximation errors. ...Therefore, the advantages of using a closed-form solution are unclear while the drawbacks are obvious. More ablation study would be helpful."
>
> We believe there is a misunderstanding. We do provide the evidence (see Figure 2) to show the inferior performance caused by the approximation errors from iterative solver. In Figure 2, we compare IGNN and IGNN-finite and illustrate the results that IGNN-finite with enough layers can outperform IGNN in synthetic experiments. The only difference between IGNN and IGNN-finite is that IGNN-finite does not use iterative solvers. Therefore, this can verify our hypothesis “the approximation errors make the iterative solver in IGNN cannot easily capture all dependencies on long chains”. In fact, this part is recognized and stated in the strong points of Reviewer emSJ. We hope our explanations can help to clear the misunderstanding.
>
> 5.	"The datasets in the experiments are very small graphs, it is unclear how the model performs in large graphs in terms of both efficiency and effectiveness. The efficiency comparison with iterative solvers will be helpful."
>
> We use these datasets as they are all used on Geom-GCN [6] and GCNII [7]. [6] uses these datasets to show that Geom-GCN’s the ability of capturing long-range dependencies and [7] shows that GCNII does not suffer from oversmoothing using these datasets. We do provide the efficiency comparison among EIGNN, IGNN, and IGNN-finite, which include the comparison with iterative solvers. We vary the size of synthetic graphs to show the efficiency comparison (see Table 1). In table 1, the graph used in the last row has 12k nodes, which we believe is not *very* small. Based on Table 1, it is clear that iterative solvers are slower than our close-form solution and IGNN-finite with many layers is even slower than IGNN which use iterative solvers.
> For additional efficiency comparison, we conduct another set of experiments on PPI dataset as in IGNN paper. PPI dataset contains multiple graphs and ~57k nodes in total. We show the resuls on PPI dataset as below:
>
> |   | Test Micro f1   |      Training time per epoch      |  Precompute time for Eigendecomposition of all S (only once) |
> |----------|----------|-------------|------|
> | EIGNN |  97.97 | 2.23s | 40.59s |
> | IGNN |    97.6   |   35.03s | N.A.|
>
> We rerun their code to get the running time. The number of epochs during training is more than 1000 as suggested in IGNN. Therefore, on PPI dataset, the total running time of EIGNN is much less than IGNN and the precompute time for eigendecomposition of S can be neglected. Meanwhile, EIGNN can achieve even slightly better performance. In addition, the experiments verify that EIGNN is applicable to multi-graph inductive setting.
>
> 6.	"In the experiments on the synthetic chain data, there is a huge performance gap between EIGNN and IGNN. Considering that IGNN is able to capture long-range dependencies, it is unclear where the improvements of EIGNN come from. In the experiments on real-world data, the performance gap between EIGNN and H2GCN is very small while the gap between EIGNN and IGNN is huge. It is a bit confusing and again raises the concern -- where do the improvements of EIGNN come from."
>
> The improvements of EIGNN come from avoiding the approximation error in IGNN. If we ignore the approximation errors, IGNN is able to capture long-range dependencies. The paper of IGNN demonstrates that IGNN can capture all dependencies on up to 10-length chains in synthetic experiments. However, we conduct comprehensive synthetic experiments to show that IGNN actually cannot easily capture long-range dependencies when the length is longer than 10 since the approximation error (see the response to point 4 for the reasons).
>
> For the experiments on real-world graphs, the graphs are all heterophilic graphs. They are suitable to test the ability of capturing long-range dependencies since a source node needs to aggregate information from farther nodes with the same class label. H2GCN directly focuses on designing GNN models for heterophilic graphs while EIGNN focuses on long-range dependencies. The better performance of EIGNN on these graphs, compared with H2GCN, shows EIGNN’s ability to capture long-range dependencies. For the reasons of why IGNN performs worse on these graphs, we believe it could be caused by the approximation error from iterative solvers as we analyse in synthetic experiment. The instabilities of iterative solvers could be also one of the reasons. In conclusion, we think that the performance improvements of EIGNN compared with IGNN can be mainly contributed to our design without iterative solvers.
>
> 7.	"While it is possible to verify the benefits of capturing long-range dependencies in the synthetic graph, it is unclear whether this property helps in real-world graphs and how to confirm it."
>
> The purpose of the synthetic experiments is to directly verify the advantage of EIGNN on capturing long-range dependencies. For the experiments on real-world graphs, we follow the setting and use the same graphs as in Geom-GCN [6], which is a previous work also focusing on long-range dependencies. Thus, we believe that the better performance achieved by EIGNN can verify the better ability of capturing long-range dependencies on real-world graphs. Only limited GNN models have this ability and some works proposing GNNs with more layers also achieve better performance (e.g., [7,8]). Therefore, we believe capturing long-range dependencies is an important property.
>
> References
>
> [1] Graph Coarsening with Neural Networks, ICLR 2021
>
> [2] GRAPH PARTITION NEURAL NETWORKS FOR SEMI-SUPERVISED CLASSIFICATION, https://arxiv.org/abs/1803.06272
>
> [3] Scaling Up Graph Neural Networks Via Graph Coarsening, KDD 2021
>
> [4] Cluster-GCN: An Efficient Algorithm for Training Deep and Large Graph Convolutional Networks, KDD 2019
>
> [5] METIS: Serial Graph Partitioning and Fill-reducing Matrix Ordering, http://glaros.dtc.umn.edu/gkhome/views/metis
>
> [6] GEOM-GCN: GEOMETRIC GRAPH CONVOLUTIONAL NETWORKS, ICLR 2020
>
> [7] Simple and Deep Graph Convolutional Networks, ICML 2020
>
> [8] Training Graph Neural Networks with 1000 Layers, ICML 2021

---

> > ### Comment · Reviewer_mzDR · 2021-08-28
> > **Discussion**
> >
> > Thank the authors for their detailed responses. The responses are helpful and they clarify some of my confusions.
> >
> > I think the paper can be further improved in the following aspects:
> >
> > 1. The description between line 126 and 128 are not sufficient to explain the motivation of the proposed model. More details are needed, for instance, why is g(F) designed as proposed.
> >
> > 2. The differences between EIGNN, IGNN, and IGNN-finite are quite confusing. It is better to provide a detailed description of these methods. Moreover, to demonstrate the benefits of EIGNN, a more extensive ablation study on the trade-off between accuracy and efficiency is necessary. For instance, the accuracy and training time should be compared with varied approximation error (iteration steps).
> >
> > 3. It will be helpful if the limitations of EIGNN can be discussed, such as whether it can be used in induction learning or dynamic graphs.
> >
> > Overall, I would like to adjust my rating from 3 to 5.

---

> > > ### Author Response · Authors · 2021-08-31
> > > **New response**
> > >
> > > Thank you for the suggestions. We post our response below.
> > >
> > > 1. We will add the motivations, which were mentioned in the previous response, to the future version of our paper. The design of $g(F)$ is to prevent divergence of the infinite sequence by making sure $g(F)$ lying within a Frobenius norm ball of radius < 1, which is also stated in our submission.
> > > 2. The differences between EIGNN and IGNN are stated in Section 4.3 (see line 193 to 198) and also the introduction. The differences between IGNN and IGNN-finite are described from line 263 to 269. We will try to provide a more detailed description in the future version.
> > > 3. We are a bit confused and not sure the exact meaning of the provided example "the accuracy and training time should be compared with varied approximation error (iteration steps)". Do you mean the training iteration steps? Or iteration steps in iterative solvers? Note that EIGNN does not require iterative solvers. For the trade-off between accuracy and efficiency, as an  example, we think we can consider how it affects accuracy and efficiency by using different k as the number of largest eigenvalues considered in truncated eigendecomposition for large graphs. Regarding this, we report some empirical results on ogbn-arxiv as blow.
> > >
> > >    | k  | Accuracy   |Training time per epoch|
> > >     |----------|----------|-------------|
> > >     | 1000 |  64.63% | 0.45s |
> > >     | 5000 |    68.53%  |   0.68s |
> > >     | 10000| 71.40% | 1.15s|
> > >
> > >     As expected, using larger k makes the accuracy increase, while the required training time per epoch increases as well. Due to limited time, we will consider more extensive studies for the future version of our paper.
> > > 4. In our submission, we mention a limitation in the conclusion. We will discuss more limitations in the future version of our paper. However, we would like to point out that EIGNN can be used under inductive learning setting. As shown in the previous response, EIGNN provides better performance compared with IGNN on PPI dataset. As we mentioned before, this experiment is indeed under the inductive learning setting. PPI dataset contains 20 training graphs, 2 validation graphs, and 2 test graphs.
> > >
> > > Hope our response can convince you and clear up your confusions. And hope you can consider raising your score if you are satisfied with our response.

---

> > > > ### Comment · Reviewer_mzDR · 2021-09-01
> > > > **Further discussion**
> > > >
> > > > Dear authors,
> > > >
> > > > Thanks for your detailed responses to my concerns.
> > > >
> > > > As far as I am concerned, the motivations you provided in the paper and your responses still lack clear justification for the design of the implicit model:
> > > >
> > > > > (1) we want residual connections; (2) learnable weights for propagation on graphs (i.e., g(F)); (3) restrict $\gamma$ to make sure the convergence of this infinite sequence.
> > > >
> > > > I believe Reviewer GtHW shares the same opinion with me:
> > > >
> > > > > W1: The design of extending SGC (from Equation 1) to EIGNN (from Equation 3) is somehow implicit and ad-hoc without clear justifications. The authors should explain this more in details for better understanding by general audiences that are not very familiar with implicit models.
> > > >
> > > > Sorry about the confusion in my previous reply. Let me restate the point on the tradeoff between accuracy and efficiency:
> > > >
> > > > I believe IGNN and IGNN-finite are both iterative solvers although they iterate in different ways. Therefore, we can reduce the approximation error using more computation and/or memory resources, such as more iteration steps in the solver. For EIGNN, the approximation error, final accuracy, and computation/memory cost will also change if you change K (top K eigenvalue) in the truncated eigendecomposition, as verified in your reply.
> > > >
> > > > In other words, for all methods including EIGNN, IGNN, and IGNN-finite, there exist tradeoffs between accuracy and efficiency. Therefore, to show the advantages of EIGNN, a detailed and comprehensive comparison of the tradeoffs for different methods are very important since efficiency is the central topic of this paper. However, in the submission and the responses, I do not see such an important study. I would recommend including more detailed studies in the next submission or the camera ready.

---

> > > > > ### Author Response · Authors · 2021-09-01
> > > > > **New response**
> > > > >
> > > > > Thank you for your suggestions and the prompt feedback.
> > > > >
> > > > > We will try to add more clear justifications for our design in the future version.
> > > > >
> > > > > Here, we share some findings regarding the tradeoffs between accuracy and efficiency. In our previous experiments, we found simply increasing the max iterative steps would not increase the accuracy of IGNN. One potential reason can be that IGNN requires one iterative solver for forward pass and another one for backward pass. Then, the approximations from these two iterative solvers are not consistent, which makes different approximation errors accumulated. Therefore, more iteration steps might not be helpful for that. We appreciate your suggestion again and we will consider adding more studies and comparisons about the tradeoffs in the future version.
> > > > >
> > > > > Additionally, we would like to clarify a misunderstanding about IGNN-finite. IGNN-finite does not require iterative solvers as well. We use IGNN-finite as ablation for IGNN to better show the limitation of IGNN (i.e., the drawback of using iterative solvers). This is also recognized by Reviewer emSJ in the strong points.
> > > > >
> > > > > We would like to emphasize again that our work focuses more on how to effectively capture long-range dependencies. EIGNN achieves this by avoiding approximation errors from iterative solvers used in previous implicit graph models.

---

### Official Review · Reviewer_GtHW · 2021-07-18

**Rating:** 7
**Confidence:** 3

**Summary:**

This paper presents an interesting idea on designing a novel graph neural network to capture the long-range dependency, which is the lack of ability of most existing works. A linear model that extends SGC with implicit infinite layers, EIGNN, is proposed to mitigate this issue. A closed form of the infinite sequences is used to define the layer-wise update rule and eigendecomposition is combined to achieve more efficient training. Experimental results on both synthetic and real-world datasets demonstrate the effectiveness of the proposed EIGNN for capturing the long-range dependency.

**Limitations And Societal Impact:**

Yes, the authors have discussed in the conclusion part.

**Main Review:**

Strengths:

S1: It is important for the GNN models to obtain the ability for capturing the long-range dependency.
S2: The idea to extend the linear model SGC into infinite layers is interesting, since it could help the community rethink the representational ability of linear models on graphs. The theoretical analysis on convergence and training of the model has merits.
S3: The paper is easy to follow with a clear motivation and comprehensive experimental analysis.

Weaknesses:

W1: The design of extending SGC (from Equation 1) to EIGNN (from Equation 3) is somehow implicit and ad-hoc without clear justifications. The authors should explain this more in details for better understanding by general audiences that not very familiar with implicit models.

W2: During the time complexity analysis, only the complexity of training is analyzed, but it seems like the computation of eigendecomposition of S, the normalized adjacency matrix with self-loops, (Line 176) is not added, which usually requires the cost of $\mathcal{O}(n^{3})$. If this is true, a full eigendecomposition of a large sparse S could make EIGNN an impractical approach for prohibiting the scalability in terms of large number of nodes $n$ for huge real-world graphs.

W3: Several concerns upon experiments include:
1)	The discussion on arbitrary hyperparameter $\gamma$ is missing, including how to set it in practice for a given graph and analyzing on the sensitivity of this hyperparameter， otherwise it will be hard for the researchers to follow.
2)	As the weakness on the analysis of complexity, why the author chooses not to evaluate the long-range dependency on the standard dataset Amazon Co-purchase as used in IGNN. Amazon Co-purchase dataset has another benefit that it can also reflect the scalability of proposed method since it is a large dataset with ~33k nodes, while the experiments on real-world dataset are all conducted on graphs that less than 10k.
3)	For the evaluation on over-smoothing, it would be interesting to see how the EIGNN performs with respect to over-smoothing under standard setting on real-world datasets, especially in comparison with variants focusing on dealing with over-smoothing, such as the setting used in GCNII.
4)	The evaluation on robustness is not very convincing since structural attack is known to be more powerful and appreciative when we attack on graph-structured data. Thus, the authors are suggested to defend their proposed model against several popular structural attack methods such as Nettack for better demonstration rather than attacks on features used in experiments.


**Time Spent Reviewing:**

~5h

---

> ### Author Response · Authors · 2021-08-10
> **Responses to all concerns raised by the reviewer**
>
> We thank reviewer GtHW for the valuable feedback and recommendations for improving the manuscript. We addressed all the concerns as described below.
>
> 1. "The design of extending SGC (from Equation 1) to EIGNN (from Equation 3) is somehow implicit and ad-hoc without clear justifications. The authors should explain this more in details for better understanding by general audiences that not very familiar with implicit models."
>
> We thank the reviewer for the valuable feedback. We list the motivation of Equation (2) – (4) as below: 1)	we want residual connections, which are also used in finite GNN models (i.e., APPNP and GCNII) and have been shown important in graph learning setting.
> 2)	Learnable weights for propagation on graphs (i.e., g(F)). In contrast, APPNP only directly propagates information without learnable weights.
> 3)	Restrict $\gamma$ to make sure the convergence of this infinite sequence.
>
> 2. "During the time complexity analysis, only the complexity of training is analyzed, but it seems like the computation of eigendecomposition of S, the normalized adjacency matrix with self-loops, (Line 176) is not added, which usually requires the cost of O(n3). If this is true, a full eigendecomposition of a large sparse S could make EIGNN an impractical approach for prohibiting the scalability in terms of large number of nodes n for huge real-world graphs."
>
> We do not add it as it is a one-time preprocessing step. For scalability problem, we can consider using the truncated eigendecomposition (only consider top k eigenvalue) for adjacency matrix S to reduce the computational and memory complexity in EIGNN. Furthermore, another possible solution to mitigate it is to use graph coarsening to reduce the size of graph or graph partition to partition a large graph into several small graphs without losing much information. We have seen several works on this topics [1,2,3]. Worth note that graph partition is also used in a well-known work Cluster-GCN [4] which focuses on scaling up GCNs to a large graph with millions nodes. Cluster-GCN splits a given graph into k non-overlapping subgraphs using METIS graph partition algorithm [5]. Therefore, we believe that graph partition is indeed a potential solution for mitigating the scalability problem in our work. How to scale up implicit graph model for large graphs is an interesting topic to explore in the future.
>
> In addition, we conduct a simple experiment on ogbn-arxiv dataset with 169k nodes. We only use top 10k eigenvalues and their corresponding eigenvectors. EIGNN can achieve 71.40% accuracy on test set. We note that the performance is not that appealing. But considering we only use 1/10 eigen vectors, we verify that truncated eigendecomposition is also a potential solution for mitigating the scalability problem. As the natural sparsity of graphs, using truncated eigendecomposition is reasonable.
>
> We also conduct another set of experiments on PPI dataset as in IGNN paper. PPI dataset contains multiple graphs and ~57k nodes in total. We conduct the experiments on it and compare the efficiency between EIGNN and IGNN. We show the resuls on PPI dataset as below:
>
> |   | Test Micro f1   |      Training time per epoch      |  Precompute time for Eigendecomposition of all S (only once) |
> |----------|----------|-------------|------|
> | EIGNN |  97.97 | 2.23s | 40.59s |
> | IGNN |    97.6   |   35.03s | N.A.|
>
> We use the test micro f1 of IGNN reported in their paper and rerun their code to get the running time. The number of epochs during training is more than 1000 as suggested in IGNN. Therefore, on PPI dataset, the total running time of EIGNN is much less than IGNN and the precompute time for eigendecomposition of S can be neglected. Meanwhile, EIGNN can achieve even slightly better performance. In addition, the experiments verify that EIGNN is applicable to multi-graph inductive setting.
>
> 3.	The discussion on arbitrary hyperparameter γ is missing, including how to set it in practice for a given graph and analyzing on the sensitivity of this hyperparameter, otherwise it will be hard for the researchers to follow.
>
> We thank the reviewer for the valuable suggestion. $\gamma$ is similar with the damping factor in APPNP. It can also be treated as the probability of aggregating the information from 1-hop neighborhood. We generally set as 0.8 in our experiments. 0.8 or 0.9 would be a good starting point to try. Here, we provide the results by varying $\gamma$ on Chameleon dataset with other best hyperparameter.
>
> | $\gamma$   | 0.1   | 0.2   | 0.3   | 0.4   | 0.5   | 0.6   | 0.7   | 0.8   | 0.9   |
> |----------|-------|-------|-------|-------|-------|-------|-------|-------|-------|
> | Accuracy | 61.45 | 62.14 | 62.45 | 62.28 | 62.80 | 63.00 | 62.96 | 62.92 | 63.00 |
>
> We can see that we should avoid the very small value (e.g., < 0.5) for $\gamma$. In general, it is not sensitive.
>
> 4.	As the weakness on the analysis of complexity, why the author chooses not to evaluate the long-range dependency on the standard dataset Amazon Co-purchase as used in IGNN. Amazon Co-purchase dataset has another benefit that it can also reflect the scalability of proposed method since it is a large dataset with ~33k nodes, while the experiments on real-world dataset are all conducted on graphs that less than 10k.
>
> We thank the reviewer for the valuable suggestion. But Amazon Co-purchase has no input features, which is rare in real world. IGNN directly use the adjacency matrix as the input features, which means the number of feature dimension would be large. EIGNN can be slow when the number of feature dimensions is large, which is a limitation of our method as mentioned in the conclusion. Thus, we did not evaluate on Amazon Co-purchase.
>
> 3.	For the evaluation on over-smoothing, it would be interesting to see how the EIGNN performs with respect to over-smoothing under standard setting on real-world datasets, especially in comparison with variants focusing on dealing with over-smoothing, such as the setting used in GCNII.
>
> We thank the reviewer for the valuable suggestion. In our work, we conduct the experiments on Chameleon, Cornell, Texas, and Wisconsin, which are all used in GCNII. For these datasets, we use the exact same setting as in GCNII. On these datasets, EIGNN consistently outperforms GCNII, which indicates that EIGNN does not suffer or suffer less from over-smoothing.
>
> 4.	The evaluation on robustness is not very convincing since structural attack is known to be more powerful and appreciative when we attack on graph-structured data. Thus, the authors are suggested to defend their proposed model against several popular structural attack methods such as Nettack for better demonstration rather than attacks on features used in experiments.
>
> We thank the reviewer for the valuable feedback and suggestions. We agree that we can further consider structural attack. We explain the reason of considering feature attack on our experiments as follows: EIGNN is an infinite-depth GNN model, which can consider more about global information and become less sensitive to perturbations from local neighborhoods. In contrast, finite message passing GNNs could be very sensitive to local neighborhoods as they usually only have a few layers and the node representations could be changed a lot when aggregating from perturbated node features. Therefore, we verify this using feature attacks.
>
> References
>
> [1] Graph Coarsening with Neural Networks, ICLR 2021
>
> [2] GRAPH PARTITION NEURAL NETWORKS FOR SEMI-SUPERVISED CLASSIFICATION, https://arxiv.org/abs/1803.06272
>
> [3] Scaling Up Graph Neural Networks Via Graph Coarsening, KDD 2021
>
> [4] Cluster-GCN: An Efficient Algorithm for Training Deep and Large Graph Convolutional Networks, KDD 2019
>
> [5] METIS: Serial Graph Partitioning and Fill-reducing Matrix Ordering, http://glaros.dtc.umn.edu/gkhome/views/metis

---

### Decision · Program_Chairs · 2021-09-27

**Decision:**

Accept (Poster)

**Comment:**

3 of 4 ratings were "accept". The lowest rating (a 5) was most concerned about insufficient experimental evaluation of speed/accuracy tradeoffs, which was echoed by other reviewers as well.

Through the rebuttal/discussion period the authors were very responsive and caused several reviewers to raise their ratings. My view is that the original submission was somewhat confusing and incomplete in a few ways, but that the core of the method and results are solid, and through the discussion period the authors have improved the paper enough to warrant publication. The confusion could also stem from the fairly high technical complexity in the work, which I'd encourage the authors to explain as clearly as possible in the final text.

I do agree with the concern about speed/accuracy analysis, and I'd encourage the authors to do anything they can to address this for camera-ready (if the paper is accepted), however I don't anticipate any surprises.